# WRN inhibition leads to its chromatin-associated degradation via the PIAS4-RNF4-p97/VCP axis

Fernando Rodríguez Pérez [1] ✉, Dean Natwick [1], Lauren Schiff [1], David McSwiggen [1], Alec Heckert[1], Melina Huey[1], Huntly Morrison[1], Mandy Loo[1], Rafael G. Miranda[1], John Filbin[1], Jose Ortega[1], Kayla Van Buren[1], Danny Murnock[1], Arnold Tao[1], Renee Butler[1], Kylie Cheng[1], William Tarvestad[1], Zhengjian Zhang [1], Eric Gonzalez[1], Rand M. Miller[1], Marcus Kelly[1], Yangzhong Tang[1], Jaclyn Ho [1], Daniel Anderson[1], Charlene Bashore[1] & Stephen Basham [1] ✉

Synthetic lethality provides an attractive strategy for developing targeted cancer therapies. For example, cancer cells with high levels of microsatellite instability (MSI-H) are dependent on the Werner (WRN) helicase for survival. However, the mechanisms that regulate WRN spatiotemporal dynamics remain poorly understood. Here, we used single-molecule tracking (SMT) in combination with a WRN inhibitor to examine WRN dynamics within the nuclei of living cancer cells. WRN inhibition traps the helicase on chromatin, requiring p97/VCP for extraction and proteasomal degradation in a MSI-H dependent manner. Using a phenotypic screen, we identify the PIAS4-RNF4 axis as the pathway responsible for WRN degradation. Finally, we show that co-inhibition of WRN and SUMOylation has an additive toxic effect in MSI-H cells and confirm the in vivo activity of WRN inhibition using an MSI-H mouse xenograft model. This work elucidates a regulatory mechanism for WRN that may facilitate identification of new therapeutic modalities, and highlights the use of SMT as a tool for drug discovery and mechanism-of-action studies.

Werner Syndrome is a rare genetic condition caused by mutations in the *WRN* gene. It is marked by accelerated aging and a predisposition to a variety of cancers[1]. The *WRN* gene encodes a RecQ helicase that plays a critical role in genomic integrity and is unique amongst RecQ helicases as it possesses an exonuclease domain in addition to its helicase domain[2]. WRN typically resides in the nucleoli of cells but undergoes DNA damage-induced translocation to the nucleoplasm to perform its repair functions, resolving a diverse array of DNA substrates, including D-loops, replication forks bubble structures, Holliday junctions, and other secondary structures[3–5]. Additionally, WRN is required for telomere maintenance, a necessary process for maintaining stem cell cellular homeostasis[6].

Synthetic lethality has emerged as an appealing approach to target cancer cells while minimizing collateral damage to otherwise healthy cells and tissue[7]. Synthetic lethality occurs when there is a simultaneous disturbance of two essential biological events leading to cell death, which would not occur in the presence of one genetic disruption alone. Highlighting the importance of WRN in DNA repair, WRN has been identified as a synthetic lethal target in microsatellite instability-high (MSI-H) tumor types that are deficient in mismatch repair (MMR) pathways[8–11]. Microsatellites are short tandem repeats of repetitive nucleotides that reside throughout the genome. Microsatellite stable (MSS) cells have two DNA repair mechanisms - (1) MMR machinery and (2) WRN to ensure genetic integrity. In MSS cells, disruption to either MMR or WRN does not lead to cell death. However, in MSI-high cells with compromised MMR processes, inhibition of WRN leads to cell death due to this synthetic lethal relationship[12]. This WRN synthetic modality can be exploited for therapeutic value and has

[1]Eikon Therapeutics, Hayward, CA 94545, USA. ✉e-mail: perezf@eikontx.com; bashams@eikontx.com

recently entered the clinic for the treatment of MSI-H cancers (NCT05838768 2018; NCT06004245 2023). Given its promise as a cancer therapeutic target, a deeper understanding of DNA replication and repair regulatory mechanisms for WRN may provide additional insights that can be exploited for therapeutic-based purposes. Here, we present the identification of a WRN degradation mechanism revealed by single-molecule tracking (SMT). Upon WRN inhibition, WRN becomes trapped on chromatin and is SUMOylated by PIAS4. SUMOylated WRN is recognized by the ubiquitin E3 ligase RNF4, which then ubiquitinates WRN, leading to p97/VCP-mediated chromatin extraction and, ultimately, proteasomal-dependent degradation, providing potential avenues for therapeutic targeting.

## Results

### The WRN inhibitor HRO761 is a potent WRN inhibitor in MSI-H cells resulting in DNA damage accumulation

To investigate the consequence of inhibition in WRN protein dynamics, we utilized a recently published clinical WRN inhibitor HRO761 (WRNi, Fig. 1a) (NCT05838768 2018)[13,14]. We first established a panel of MSI-H and MSS cells and showed that depleting WRN by siRNA results in the induction of γH2AX, a marker of DNA damage, exclusively in the MSI-H cell lines (Supplementary Fig. 1a–d). We then tested these same cell lines with the WRN inhibitor HR0761 and observed a > 100-fold induction of DNA damage response in MSI-H cells compared to MSS cells, resulting in apoptosis and cell death (Fig. 1b–d and Supplementary Fig. 1g). Additionally, the antiproliferative effect was only observed in MSI-H, but not MSS cells, further highlighting the specificity of WRNi (Fig. 1e and Supplementary Fig. 1h, i). To profile the specificity of this compound in vitro, we purified the WRN protein and its related homolog, BLM (Supplementary Fig. 1e, f). Using this recombinant system, we observed a >1000-fold difference in ATPase and helicase inhibition with WRN vs BLM (Fig. 1f, g and Supplementary Fig. 1j, k). Having established this system, we also profiled the putative WRN inhibitor NSC-617145, which has been reported to inhibit WRN biochemical activity. However, we saw no activity of this compound in any of our optimized biochemical or cell-based assays (Supplementary Fig. 1l). Furthermore, other groups have also reported that any activity observed with these compounds comes from assay artifacts and interference (Heuser et al., 2024). Therefore, no additional profiling of this compound was pursued.

### Single-molecule tracking (SMT) can measure the trapping of WRN on chromatin upon its inhibition

To enable imaging of proteins at the single-molecule level, we used CRISPR to generate endogenous HaloTag™ WRN fusion protein (WRN^Halo) in MSS (U2OS) and MSI-H (HCT-116) cells[15]. Prior to our imaging studies, we validated the successful tagging of WRN in these cells by depleting the HaloTag™ WRN signal using a HaloTag™ PRO-TAC (Supplementary Fig. 2c, d). Furthermore, a comparison between WT and HaloTag™-WRN HCT-116 cells after WRN inhibition showed no discernable difference in γH2AX induction between the two cell lines (Supplementary Fig. 2e).

WRN undergoes a sub-compartmental translocation in response to DNA damage which can be viewed using standard immunofluorescence[5,16]. As previously reported, we observed robust translocation of WRN from the nucleolus to the nucleoplasm upon induction of DNA damage (Fig. 2a)[17,18]. To better understand the link between WRN enzymatic activity and its mobility in cells, we next used WRNi in combination with SMT to visualize the dynamics of individual WRN molecules in the nuclei of live cells. Our fully automated SMT platform enables the measurement of thousands of experimental conditions and millions of cells per day[15]. Recently we introduced substantial improvements to the platform, utilizing a light-sheet-based illumination strategy that improves both the throughput and quality of the SMT platform[19]. After image acquisition, SMT movies are

processed using a custom data analysis pipeline that first identifies the positions of fluorescent emitters with a statistical test (Supplementary Fig. 3a–d). After identification, fluorescent emitters are localized to subpixel precision and spatiotemporally linked into trajectories (Supplementary Fig. 3e–j). These trajectories can provide insights into the residence time and mobility of proteins of interest (Supplementary Figs. 3k, 4).

Upon WRNi treatment, we observed a significant and dose-dependent decrease in average WRN mobility in the HCT-116 background (Fig. 2b–d, Supplementary Fig. 2f, and Supplementary Movie 1). We observed no changes in protein diffusion coefficient in U2OS cells, suggesting that the microsatellite state of the cell influences the effect of WRNi on WRN dynamics (Fig. 2c and Supplementary Fig. 2g). Due to the fact that SMT is a single-molecule assay, we next sought to extract a more granular view of dynamic states of WRN in cells. To this end, we plotted the proportion of WRN proteins moving within a range of diffusion rates to generate a state-array histogram that provides a glimpse of the various states in which WRN is found, ranging from an immobile chromatin-bound state to a freely diffusing state[15]. We observed a decrease in the fastest-diffusing states and more than a twofold increase in the chromatin-bound fraction of WRN in HCT-116 cells upon WRNi treatment (Fig. 2e–g). In contrast, we did not observe these changes in microsatellite stable U2OS cells or with the treatment of general DNA-damaging compounds (Fig. 2c and Supplementary Fig. 2g). The observed decrease in WRN diffusion coefficient was subsequently followed by a decrease in SMT spot density in the nucleus, suggesting possible degradation of the WRN protein (Fig. 2h).

We confirmed this observation of WRN degradation by performing a time course of WRNi which resulted in a reduction of WRN protein levels in HCT-116 cells but not U2OS cells, corroborating the SMT findings (Fig. 3a, c and Supplementary Fig. 5a, b). We were not able to see WRN degradation in other MSS cell lines, further suggesting that WRN degradation upon its inhibition by HRO761 is MSI-H dependent (Supplementary Fig. 6a). WRN degradation does not result from general DNA damage, but instead is specific to WRNi in a dose-dependent and proteasomal-dependent way, as co-treatment of HCT-116 with the proteasome inhibitor carfilzomib (CFZ) rescues this effect (Fig. 3k and Supplementary Fig. 5b–d). Using cycloheximide, we determined that WRN inhibition reduced the half-life of WRN protein by over an order of magnitude, from 16.6 to 1.5 h (Fig. 3b and Supplementary Fig. 5e). Taken together, these data support our initial observation of WRN degradation via SMT and indicate that treatment with WRNi results in the proteasomal-dependent degradation of WRN protein.

### WRN inhibition leads to its trapping and subsequent proteasomal degradation in a VCP/p97-dependent manner

The increased proportion of chromatin-bound (i.e., bound) WRN molecules, and the concomitant decrease in the freely diffusing (i.e., fast) population shown in the state-array data, suggests that inhibition of WRN by HRO761 leads to its trapping on DNA followed by what we hypothesize to be the degradation of WRN (Fig. 2e–h). This mode of action, chromatin-associated degradation, has been widely studied with PARP1 and its inhibitors[20,21].

To investigate whether a similar mechanism of action occurs upon WRN inhibition, we treated WRN^Halo cells with WRNi followed by permeabilization with a mild detergent to liberate soluble contents of the cells followed by fixation for immunolocalization studies[22]. These solubilization experiments revealed that treatment with WRNi led to the accumulation of WRN bound to chromatin based on the retention of WRN signal detected in the nucleus after treatment with detergent (Fig. 3d and Supplementary Fig. 6b). Strikingly, we also observed increased colocalization of ubiquitin and WRN upon treatment with WRNi, further fortifying the link between WRN inhibition and the ubiquitin-proteasome pathway (Fig. 3d, e and Supplementary Fig. 6f). This link was also shown by using tandem ubiquitin-binding entities

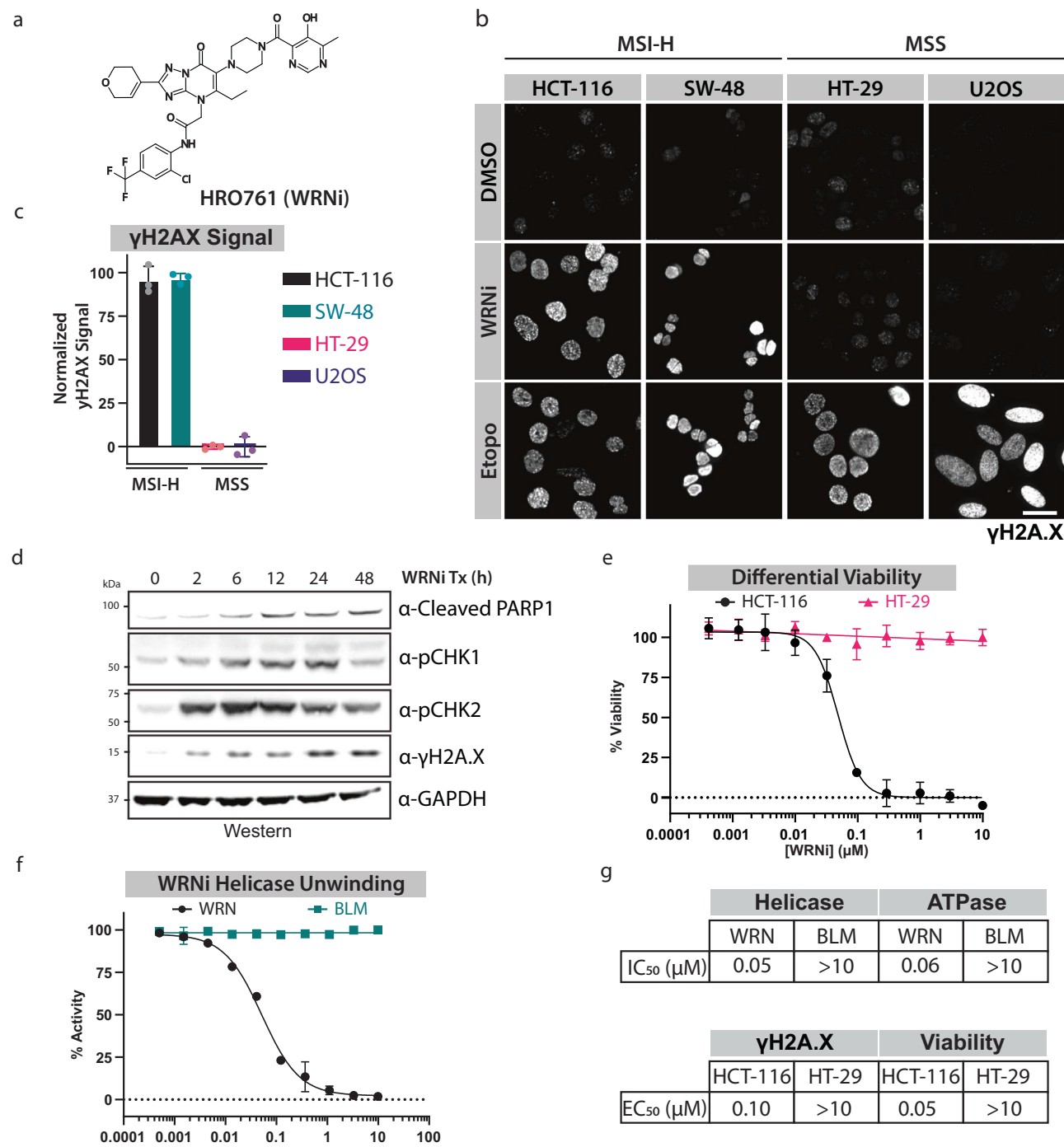

**Fig. 1 | HRO761 is a specific and potent WRN inhibitor. a** Chemical structure of HRO761 (WRNi). **b** WRN inhibition leads to induction of the DNA damage response in MSI-H cells. Phospho-histone H2A.X (Ser139) (γH2A.X) staining was used to visualize DNA damage in MSI-H (HCT-116, SW-48) and MSS (HT-29, U2OS) cells after 24 h treatment with indicated compounds. Scale bar = 20 μm. **c** Quantification of γH2A.X signal levels in cells in (**b**), normalized to DMSO and Etoposide. Graphs represent averages of $n = 3$ plate replicates. **d** WRN inhibition by WRNi leads to an induction of the DNA damage response resulting in apoptosis. HCT-116 cells were treated with 10 μM WRNi and analyzed by Western Blot (WB) to assess DNA damage response markers. GAPDH was used as a loading control. **e** Dose-response curves are measuring the viability of HCT-116 cells or HT-29 cells after treatment with WRNi for 4 days. Graphs represent averages from $n = 6$ plates. **f** Dose-response curves measuring in vitro WRN or BLM unwinding activity after treatment with WRNi. Graphs represent averages from $n = 4$ replicates. WRN and BLM unwinding activity is normalized to DMSO. **g** Summary of $IC_{50}$ or $EC_{50}$ values of WRNi in the indicated biochemical and cellular assays. All curve fits were done by fitting a 4-parameter logarithmic regression curve. All error bars represent standard deviation (s.d.). DMSO is dimethyl sulfoxide; BLMi is BLM inhibitor Compound 2; Etopo is etoposide. MW is molecular weight. Source data are provided as a Source Data file.

(TUBEs), which revealed a clear enrichment of higher molecular weight WRN species after WRNi treatment (Fig. 3f).

We further validated these findings by performing cellular fractionations, separating soluble cellular contents from chromatin (Supplementary Fig. 5g). At steady state, WRN is tightly bound to chromatin, remaining mostly bound to chromatin in 150 mM NaCl CSK buffer (Supplementary Fig. 5h). Performing cellular fractionations in buffers with different ionic strengths, we observed that WRN

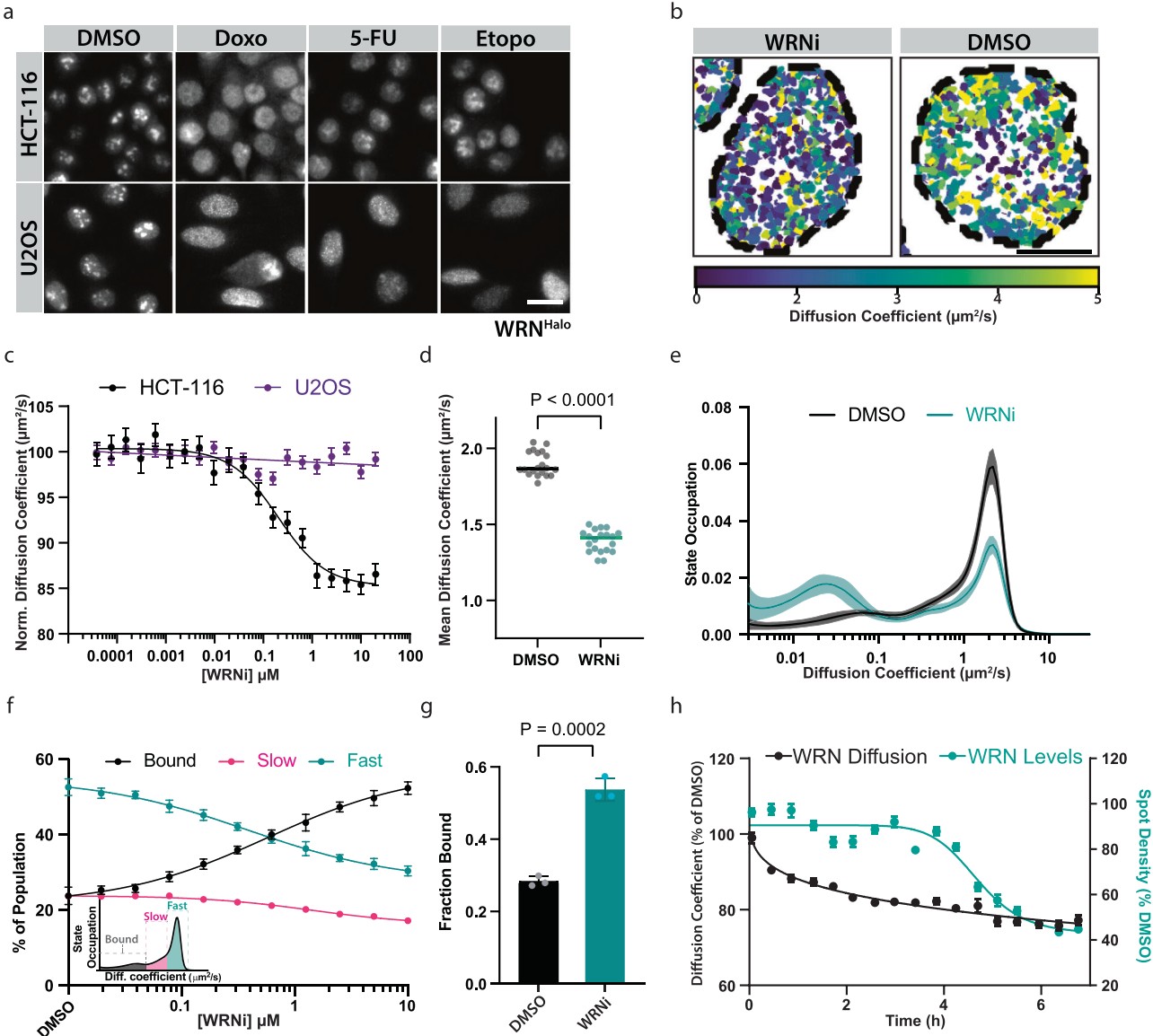

**Fig. 2 | Single-molecule tracking shows a change in WRN cellular dynamics in an MSI-H-dependent manner. a** Validation of WRN^Halo cell lines, HCT-116 and U2OS. Subcellular localization of WRN in the presence or absence of DNA-damaging compounds. WRN^Halo successfully translocates across compartments, suggesting a functional protein. Scale bar = 20 μm. **b** Representative SMT tracks overlaid with Hoechst nuclear stain outlines, in the presence or absence of 10 μM WRNi. Each track is colored according to the diffusion coefficient of the molecule it represents. **c** Inhibition of WRN only affects its mean diffusion coefficient in MSI-H cells. Dose-response curves with WRNi measuring the diffusion coefficient of WRN^Halo after 4 h treatments in HCT-116^WRN-Halo or U2OS^WRN-Halo. The graph represents the average from $n = 4$ plates per condition. Error bars represent the standard error of the mean (s.e.m.). **d** Dot plot quantification of WRN diffusion coefficient after treatment with 10 μM WRNi. Each point represents the average WRN diffusion coefficient across all the nuclei in each FOV. $n = 20$ plates. Lines represent sample medians. $P$ values were calculated using a two-tailed, unpaired $t$-test **e** WRNi shifts a large fraction of

molecules from the free-diffusing state ("fast") to the chromatin-bound ("bound") state. Distribution of diffusive states in HCT-116^WRN-Halo cells showing the relative proportion of WRN molecules as a function of diffusion coefficient occupation, in the presence and absence of 10 μM WRNi. Plot line represents sample means, shaded area represents s.d. **f** Dose-response curves with WRNi measuring the different diffusive states of WRN protein. Inset is a representation of how diffusive states are classified. Error bars represent s.d., $n = 8$ plates. **g** Quantification from (**f**) of the chromatin-bound fraction of WRN^Halo protein in the presence or absence of 10 μM WRNi. Error bars represent s.d., $n = 3$ plates. **h** Overlay of WRN^Halo diffusion coefficient and molecule spot densities after treatment with 10 μM WRNi over the indicated timepoints. Values are normalized to DMSO. $n = 2$ plates. Error bars represent s.e.m. All $P$ values were calculated using a two-tailed, unpaired Student's $t$-test. DMSO is dimethyl sulfoxide; WRNi is HRO761; 5-FU is 5-fluorouracil, Doxo is doxorubicin. Source data are provided as a Source Data file.

dissociates from chromatin at 300 mM NaCl (Fig. 3g and Supplementary Fig. 5i). However, treatment with WRNi caused WRN to remain bound to chromatin under these conditions, in a dose and time-dependent manner (Fig. 3g–i and Supplementary Fig. 5i). WRN dissociated from chromatin at high salt concentrations (500 mM), suggesting that the chromatin trapping of WRN is not covalently linked (Fig. 3g). Taken together, these data indicate that upon inhibition, WRN is trapped on chromatin and subsequently ubiquitylated.

Consistent with the lack of γH2A.X induction observed in U2OS cells upon WRN inhibition, WRNi treatment in this MSS background did not result in WRN trapping or degradation, suggesting that microsatellite instability plays a role in the regulation of WRN upon its inhibition (Fig. 3j and Supplementary Fig. 6c).

DNA replication machinery has the potential to become stalled on DNA due to blockade by other bound proteins or DNA lesions[23,24]. To mitigate trapping, cells take advantage of ubiquitin-mediated

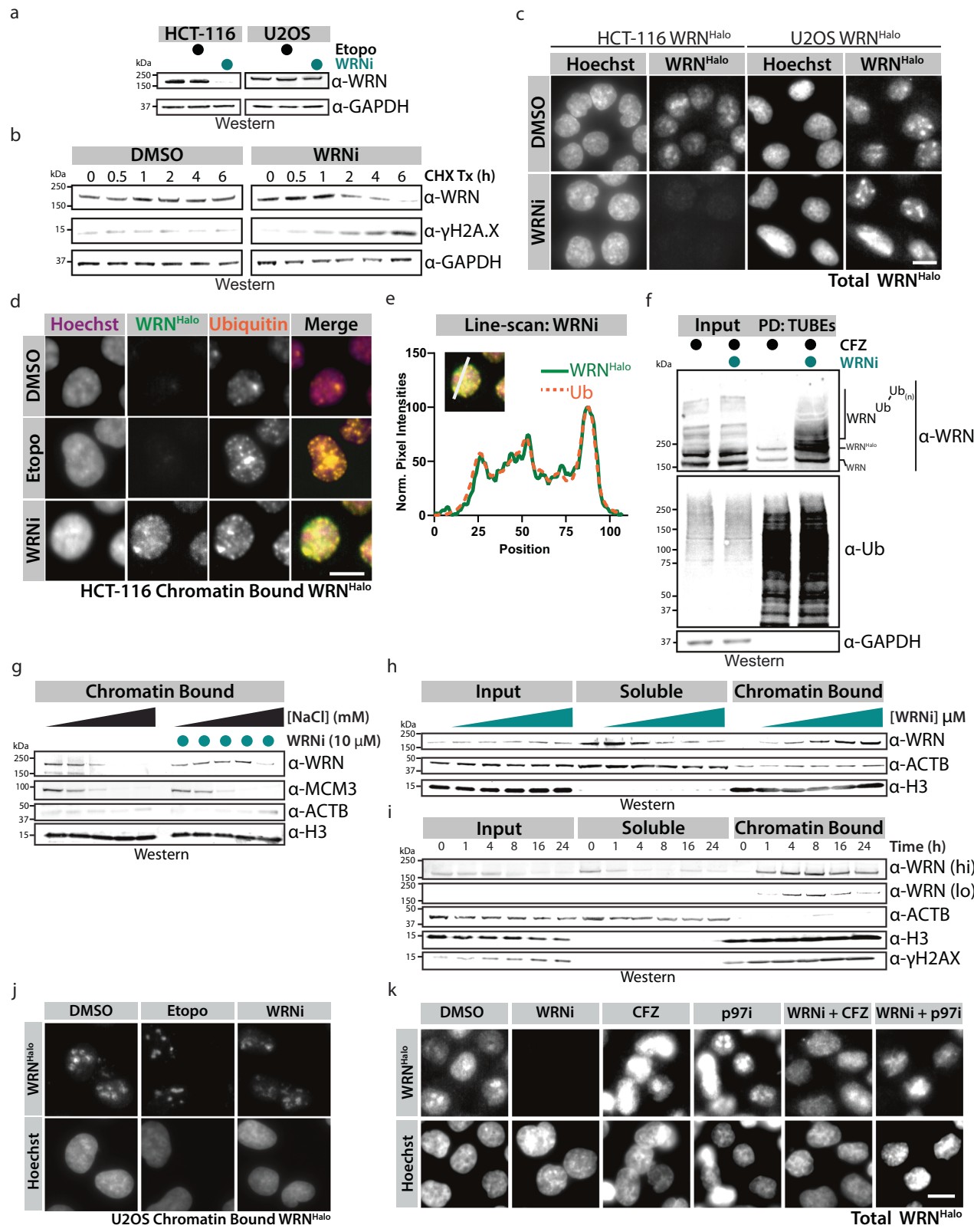

proteasomal degradation in which these stalled proteins are marked for degradation by classical proteasome pathways after ubiquitin deposition[25]. p97/VCP has been implicated in the extraction of ubiquitylated membrane-bound, and chromatin-bound proteins. Therefore, we next investigated if the p97/VCP-proteasome axis was responsible for the degradation of WRN[26–29]. Indeed, upon co-treatment with p97/VCP or proteasome inhibitors, we were able to

rescue the degradation of WRN induced by WRNi in HCT-116 cells, suggesting an endogenous ubiquitin-dependent regulatory paradigm for WRN (Fig. 3k and Supplementary Fig. 6d). This finding was further validated using cellular SMT (Supplementary Fig. 6e)[30,31]. We observed that after 16 h of WRNi treatment, chromatin-bound levels of WRN decreased (Fig. 3i). We reasoned that this decrease in WRN levels was due to proteasomal-dependent turnover which is consistent with our

**Fig. 3 | WRN Inhibition leads to its chromatin-associated degradation. a** WB analysis of HCT-116 or U2OS cells treated with 10 µM WRNi or etoposide for 16 h. **b** Treatment of HCT-116 cells with 100 µg/mL of cycloheximide (CHX) in the presence or absence of 10 µM WRNi, followed by WB analysis. Quantifications of CHX chase are in Supplementary Fig. 5b. **c** Loss of WRN protein after treatment with WRNi can be visualized by microscopy. HCT-116[WRN-Halo] or U2OS[WRN-Halo] were treated with 10 µM WRNi for 16 h and imaged. Quantification is in Supplementary Fig. 5c. Scale bar = 10 µm. **d** WRN inhibition leads to WRN trapping on chromatin and its ubiquitylation. HCT-116[WRN-Halo] cells were treated with 10 µM WRNi or etoposide for 6 h, followed by imaging. Trapped WRN quantification is in Supplementary Fig. 5b. Scale bar = 10 µm. **e** Line-scan quantification of WRNi treated cells from (**d**). Line-scan quantifications of etoposide treatments are in Supplementary Fig. 5f. **f** WRN inhibition leads to its ubiquitylation. Tandem ubiquitin-binding entities (TUBEs) pulldown (PD) of HCT-116 cells after treatment with 10 µM WRNi for 6 h in the presence of 1 µM carfilzomib (CFZ), and subsequent blotting with indicated antibodies. **g** WB analysis of chromatin fractionations of HCT-116 cells in the presence or absence of WRNi treated for 2 h, performed under various NaCl concentrations (25, 75, 150, 300, or 500 mM NaCl). Inputs and soluble fractions in Supplementary Fig. 5i. **h** Subcellular fractionations of HCT-116 cells as in (**g**), performed under 300 mM CSK buffer with increasing concentrations of WRNi (0, 0.03, 0.1, 0.3, 1.0, 3.0 µM). **i** WB analysis of HCT-116 cells subjected to a 10 µM WRNi time course, followed by subcellular fractionation as in (**h**). "hi" indicates a high WB exposure, and "lo" a low exposure. **j** WRN chromatin trapping upon its inhibition is MSI-H dependent. Cells were treated with WRNi as in (**d**) but using the MSS cell line U2OS[WRN-Halo]. Trapped WRN Quantification is in Supplementary Fig. 5e. **k** Degradation of WRN is dependent on the p97/VCP-proteasome axis. HCT-116[WRN-Halo] were treated with 10 µM of WRNi and 1 µM of either CB-5083 (p97i) or CFZ for 6 h, then imaged. Quantifications are in Supplementary Fig. 5c. DMSO is dimethyl sulfoxide; WRNi is HRO761; CHX is cycloheximide; CFZ is carfilzomib; Etopo is etoposide. All *P* values were calculated using a two-tailed, unpaired Student's *t*-test. ns not significant. MW is molecular weight. Source data are provided as a Source Data file. For WBs in **a**, **b**, **f** GAPDH was used as a loading control; for (**g–i**), ACTB and H3 were used as processing controls.

data from time course experiments in which we see WRN protein levels decrease after 8 h of WRNi (Fig. 3i and Supplementary Figs. 5a, 6f). These findings were corroborated by performing cellular fractionations in the presence of a p97/VCP inhibitor, which lead to the rescue of WRN degradation after WRNi treatment, and persistent chromatin-bound WRN (Supplementary Fig. 6f).

These data demonstrate that SMT can be used as a robust and rapid technique to measure the binding states of chromatin-associated proteins such as WRN. These chromatin fractionation experiments validate our SMT results showing that WRNi leads to chromatin trapping of WRN, which results in its degradation in a p97/VCP dependent manner.

## A phenotypic screen revealed the ubiquitin ligase RNF4 as responsible for the degradation of trapped WRN

Having determined that WRN degradation by WRNi treatment is mediated by the p97/VCP-proteasome axis, we next set out to identify the ubiquitin E3 ligase that activates this process. Depletion of the E3 ligases MIB1 and MDM2, which have been reported to regulate WRN, did not prevent the degradation of WRN protein upon WRN inhibition (Supplementary Fig. 7a)[32,33]. Therefore, we performed a ubiquitome-focused phenotypic siRNA screen and identified six potential genes that rescue WRNi-mediated degradation of WRN or regulate WRN protein levels in other ways (Fig. 4a and Supplementary Fig. 7b, c). Further validation of these siRNA hits identified RNF4 as the ligase responsible for WRN degradation upon WRNi treatment (Fig. 4a, b). Knock-down of RNF4 showed the most significant rescue of the phenotype, with all RNF4 siRNA oligos rescuing WRN protein levels to at least 75% of control-treated cells (Supplementary Fig. 7c). We speculate that the partial rescues from the other ubiquitin modulators indicate putative genetic interactors, but the lack of a full rescue effect suggests these are not direct modulators, such as UBE2D3, which is a known E2 conjugating enzyme for RNF4 (Fig. 4c and Supplementary Fig. 7c)[34,35]. We also observed a rescue of the WRN degradation phenotype using SMT by measuring the density of labeled WRN molecules in the nucleus after RNF4 siRNA knock-down (Supplementary Fig. 7d). Depletion of RNF4 abolished the ubiquitylation of WRN after WRNi treatment, further establishing that RNF4 is responsible for WRN degradation (Fig. 4d–f and Supplementary Fig. 6e). Depletion of RNF4 only slightly rescued the change in WRN diffusion coefficient after inhibition by WRNi, suggesting that WRN remains trapped on chromatin in the absence of ubiquitylation (Supplementary Fig. 7f).

To further confirm that RNF4 was the ligase responsible for the degradation of WRN upon WRNi treatment, we set up an inducible stable expression system to functionally validate this phenotype (Supplementary Fig. 8a) (Rodriguez-Perez et al., 2021). Using this inducible system, we can selectively re-introduce wild-type or an E3 ligase-dead mutant (C132S/C135S) of RNF4 that are resistant to the

RNF4 siRNAs used in this study (Supplementary Fig. 8b)[36]. Using this system, we first confirmed our original observation that WRN degradation is rescued by the depletion of wild-type RNF4 (Fig. 4g, i). However, we fail to see a rescue of the degradation phenotype when RNF4[C132S/C135S] is expressed, suggesting that the ubiquitin ligase activity of RNF4 is indeed required for the degradation of WRN upon WRN inhibition (Fig. 4h, i). Having demonstrated that RNF4 is necessary for WRN ubiquitylation and degradation, we next interrogated whether this was a direct result of its ubiquitin ligase activity, or through a different mode of action, such as its transcriptional regulation[37,38]. To achieve this, we performed a TUBE pulldown experiment and show that in the absence of endogenous RNF4 but with stably overexpressed RNF4[WT], we see ubiquitylation of WRN with WRNi treatment (Fig. 4j). This ubiquitylation is absent in the presence of RNF4[C132S/C135S], suggesting that the E3 ligase activity of RNF4 is required for WRN ubiquitylation after WRN inhibition (Fig. 4j). Failure to remove trapped proteins from chromatin can be detrimental for cellular homeostasis (Hopkins et al., 2019; Krastev et al., 2022). This led us to hypothesize that trapped WRN would be detrimental to genome integrity, and failure to remove and degrade trapped WRN results in further DNA damage. Indeed, this was evident by the observed increased γH2A.X accumulation following the knockdown of RNF4 (Supplementary Fig. 8c).

To further probe the pathway, we performed treatments with the E1 ubiquitin-activating enzyme inhibitor TAK-234 (E1i). Treatment with E1i and WRNi led to a slight decrease in the diffusion coefficient of WRN, which could suggest that inhibited WRN remains bound to chromatin if it cannot be ubiquitylated[39] (Supplementary Fig. 8d). State-array analysis revealed that the global inhibition of ubiquitylation by E1i alone led to an increase in the slow diffusing fraction of WRN protein (Fig. 4k and Supplementary Fig. 8e). This indicates that ubiquitylation is indeed required for the regulation of WRN beyond its function to drive protein degradation as has been shown for various cellular processes[40–42]. However, the state-array analysis also revealed that co-treatment with E1i and WRNi led to a slight but significant increase in the chromatin-bound fraction of WRN compared to WRNi treatment alone (Fig. 4k and Supplementary Fig. 8f). Taken together, these data suggest that WRN inhibition in MSI-H cells leads to an increase in chromatin binding, followed by ubiquitylation and subsequent degradation mediated by RNF4. This ubiquitylation is necessary to remove WRN from chromatin, as exemplified by the increase in WRN bound to chromatin in the presence of an E1 inhibitor.

## The PIAS4-RNF4-p97/VCP axis is responsible for the ubiquitylation and degradation of trapped WRN

RNF4 is known as a SUMO-targeted ubiquitin ligase (STUBL) and has been reported to regulate trapped chromatin proteins such as PARP1,

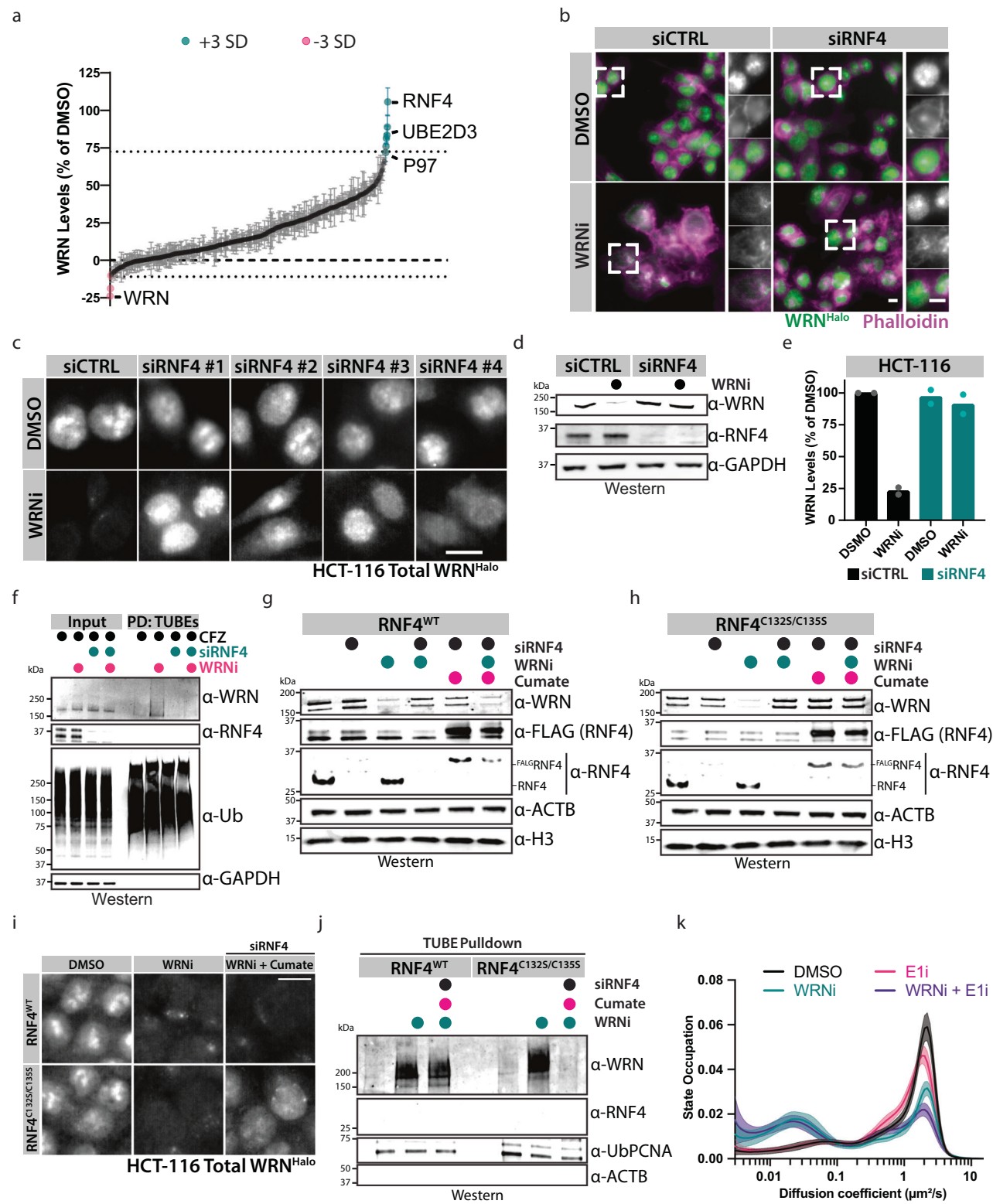

DNMT1, and TOP1A[43–45]. To investigate if SUMOylation is required for the degradation of inhibited WRN, we depleted cells of the SUMO ligase PIAS4. The depletion of PIAS4 provided a modest, yet significant rescue of WRN degradation upon WRNi treatment (Fig. 5a, c). We posit that this failure to completely rescue the WRN degradation phenotype is due to redundancies among the different PIAS proteins and moderate depletion after siRNA treatment (Supplementary Fig. 8g). To investigate this, we co-treated cells with the SUMO E1 activating enzyme inhibitor ML-792 (SUMOi) and WRNi[46]. Treatment with SUMOi

led to a full rescue of the degradation phenotype by various means of detection, including SMT (Fig. 5b–f and Supplementary Fig. 9a, b). Furthermore, upon SUMOylation inhibition, we failed to pull down ubiquitylated WRN even in the presence of WRNi, indicating that the SUMO cascade is necessary to initiate ubiquitylation of WRN (Fig. 5g). This is further exemplified by the appearance of higher molecular weight WRN, which corresponds to ubiquitylated and SUMOylated WRN, as they disappear upon co-treatment with SUMOi (Fig. 5h). Co-treatment with SUMOi and WRNi lead to a further decrease in WRN

**Fig. 4 | Phenotypic siRNA screen identified RNF4 as the ubiquitin E3 ligase targeting WRN for degradation. a** A phenotypic screen identified genes involved in WRN degradation. Colored circles indicate hits that are 3 standard deviations (SD) from the mean; error bars represent s.d., $n = 2$ biological replicates. Quantification of WRN protein levels after treatment with siRNAs in the presence or absence of 10 µM WRNi for 24 h. **b** Representative images of HCT-116^WRN-Halo cells from the siRNA screen performed in (**a**). **c** siRNA SMARTpool decomplexification and validation. Representative images of HCT-116^WRN-Halo treated with 10 uM WRNi. Quantifications are in Supplementary Fig. 7c. **d** WB analysis of HCT-116 cells treated with siRNF4 oligos for 24 h, then treated with or without 10 µM WRNi for an additional 24 h. **e** Quantifications of (**d**). Graphs represent the mean value of $n = 2$ biological replicates. **f** WB analysis of TUBES pulldowns (PD) after depletion of RNF4. HCT-116 cells were treated with siRNF4 oligos for 24 h, then treated with or without 10 µM WRNi for 6 h. All samples were treated with CFZ. **g** WB analysis of HCT-116^WRN-Halo;FLAG-RNF4 cells treated with 10 µM WRNi after depletion of RNF4,

following induction of stably integrated RNF4. **h** WB analysis as in (**g**) but using an RNF4^Cl32S/Cl35S catalytic mutant. **i** Samples were treated as in (**g**) with the addition of JF549, and imaged. Images are representative of $n = 3$ biological replicates **j** WB analysis of HCT-116^WRN-Halo under the indicated drug treatments and induction of indicated RNF4 constructs. All samples are treated with CFZ. Inputs and total ubiquitin loading controls are in Supplementary Fig. 7g. Ub-PCNA was used as a processing control. **k** Distribution of diffusive states in HCT-116^WRN-Halo cells showing the relative proportion of WRN molecules as a function of diffusion coefficient occupation after treatment with 1 µM E1i in the presence or absence of 10 µM WRNi. Plot line represents sample means, shaded area represents s.d. DMSO is dimethyl sulfoxide; WRNi is HRO761; E1i is TAK-243; CFZ is carfilzomib. MW is molecular weight. All scale bars = 10 µm. WB data shown in this figure are representative images of biological duplicates ($n = 2$). Source data are provided as a Source Data file. For WBs in **d**, **f** GAPDH was used as a loading control; for **g**, **h** ACTB and H3 were used as loading controls. In **f**, UB was used as a processing control.

---

diffusion coefficient (Fig. 5i). State-array analysis indicated that this decrease was due primarily to a decrease in the mobile fraction of WRN, further suggesting that the SUMO/ubiquitin cascade is important for the regulation of inhibited WRN (Fig. 5j). Indeed, inhibition of the SUMO-Ubiquitin-p97/VCP cascade led to an increase in chromatin-bound WRN following inhibition by WRNi on our detergent extraction experiment (Supplementary Fig. 9c, d). Taken together, these data show that small molecule inhibition of WRN leads to its trapping onto chromatin, leading to its SUMOylation and subsequent ubiquitylation by RNF4 which ultimately results in proteasomal degradation.

## WRN inhibition degrades WRN and has potent antitumor activity in a mice HCT-116 xenograft model

Having shown that WRNi leads to the robust and specific inhibition of WRN in a recombinant protein system and tissue culture model, we wanted to interrogate whether WRNi had antitumor effects in vivo. To this end, we assessed the antitumor activity of WRNi in nude mice implanted with HCT-116 xenografts. WRNi was administered orally, at daily doses of 15, 30, or 90 mg/kg, and tumor volume was measured twice per week. All doses administered showed robust tumor inhibition, with tumor growth inhibitions of 59%, 89%, and >100% for the 15, 30, and 90 mg/kg groups, respectively (Fig. 6a). All dosing groups tolerated the compound, with no changes in body weight or other clinical signs observed for all mice (Fig. 6b). To assess the in vivo target engagement of the compound, tumor tissue was dissected from mice following the efficacy study. Given that in vitro treatment of WRNi in HCT-116 cells led to WRN degradation, we decided to measure WRN protein levels from tumor samples. Indeed, WRN degradation was observed in the tumors from all dose groups, suggesting that WRNi achieved in vivo target engagement in an HCT-116 xenograft tumor model in mice (Fig. 6c, d).

Taken together, these data show that WRN inhibition in vivo results in potent antitumor effects and is well tolerated by mice. Furthermore, it appears that the mechanism of action is recapitulated in mice, as evidenced by WRN degradation in tumor samples after treatment with WRNi.

## Discussion

This work has elucidated a molecular pathway that regulates WRN activity upon its inhibition. Using a super-resolution microscopy platform, we have shown that SMT can be used at scale to dissect the molecular mechanism of actions of drugs, allowing for the rapid and robust identification and quantification of protein behavior that standard approaches may overlook. This technology greatly reduces the time required to measure small molecule effects on targets, accelerating drug discovery and research. The identification of WRN as a synthetic lethal target in MSI-H cancers has opened new avenues for therapeutic intervention of these malignancies. Several approved clinical drugs selectively kill cancer cells by inhibiting DNA-binding

proteins and conferring synthetic lethality[47,48]. For example, inhibition of the enzyme Poly ADP-ribose polymerase (PARP) can induce specific cell toxicity by PARP trapping. PARP carries out its enzymatic function by repairing single-stranded DNA breaks, but inhibition leads to trapping of PARP on DNA, accumulation of DNA damage, stalled replication forks, and cytotoxicity[49–51].

Using a combination of SMT, microscopy, and classical biochemical methods, we have shown that WRN inhibition leads to its chromatin trapping and degradation in an MSI-H-dependent way. Using a phenotypic screen, we then identified the ubiquitin ligase RNF4 as being responsible for the degradation of chromatin-trapped WRN. Trapped WRN is recognized by the SUMO ligase PIAS4, through a yet-to-be-discovered mechanism, which then serves as a recruitment signal for RNF4. This cascade culminates in the extraction of WRN through p97/VCP, resulting in proteasomal-dependent degradation of WRN (Fig. 6e). WRN that cannot be released from chromatin after trapping is detrimental and leads to persistent DNA damage, even after relief of WRN inhibition (Fig. 5l and Supplementary Fig. 10a, b). Although we have expanded our understanding of WRN biology by elucidating a WRN regulatory pathway, much remains unknown. The observation that WRN is only degraded in MSI-H cells warrants further investigation. We hypothesize that for chromatin trapping of WRN to occur, WRN must be actively surveilling DNA. This would only occur in the context of a high degree of microsatellite instability. The absence of such microsatellites in MSS cells would mean that WRN is not actively bound to DNA, sparing it from degradation upon its inhibition.

This regulatory cascade has various points at which potential therapeutic interventions are possible, such as co-inhibition of the SUMOylation cascade or depletion of RNF4. To test this hypothesis, we performed genetic manipulations to deplete PIAS4 or RNF4 in the presence or absence of WRNi, and subsequently measure cell viability. In line with our previous observations which saw persistent DNA damage after inhibition of the SUMO/Ubiquitin pathway, we saw a synergistic shift in the $EC_{50}$ of WRNi after depletion of RNF4 (Supplementary Fig. 10c). Depletion of PIAS4 led to a profound viability defect even in the absence of WRNi. We reason this is due to SUMOylation being a critical process for the maintenance of DNA homeostasis, and overall cell health, making chronic depletion of the SUMO ligase PIAS4 is not well tolerated by cells. To circumvent this, we used a small molecule inhibitor (SUMOi) in combination with WRNi. This treatment results in a synergistic shift in the $EC_{50}$ of WRNi (Fig. 5k and Supplementary Fig. 10d–h)[52]. Many other areas of potential intervention exist, such as disruption of the translocation of WRN from the nucleolus to the nucleoplasm, or perturbation of the molecular players that recognize chromatin-trapped WRN protein; an area that requires further studies to elucidate those players. This work provides deeper insights into the regulation of WRN protein activity upon small molecule inhibition that was gained in part through super-resolution microscopy techniques like cellular SMT. Such detailed understanding

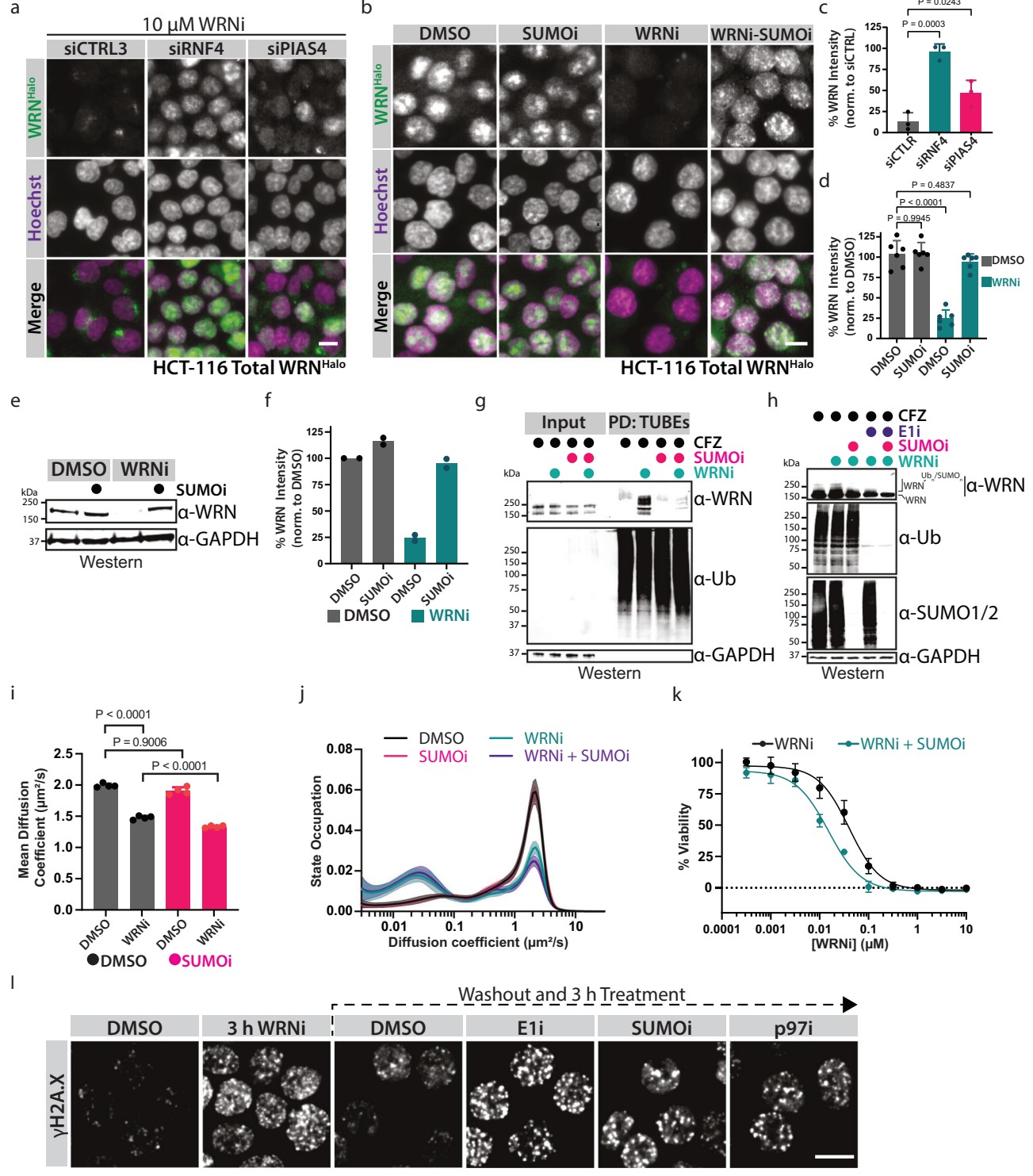

will likely be critical in defining optimal clinical strategies as first-in-class WRN inhibitors undergo clinical testing. Furthermore, our work underscores how new technologies like SMT can be used to elucidate the mechanistic underpinnings of small molecules, and may facilitate the discovery of new therapeutic agents.

## Methods
### Ethics regulation statement
All mice studies were performed at the Mispro Vivarium facility in New York, NY and were conducted according to the guidelines of the Mispro Institutional Animal Care and Use Committee and Eikon Therapeutics protocol. The IACUC protocol for these studies is EIK-01.

### Statistics and reproducibility
Statistical tests used throughout the manuscript are detailed in the figure legends. All statistical analyses were performed using Prism v10.

### Tissue culture
HCT-116 (MSI genotype), U2OS, RKO, SW-48, DLD-1, and HT-29 (MSS genotype) cells were used. Cells were cultured in Gibco McCoy's 5A Medium (Thermo, 16600082) containing 10% fetal bovine serum (FBS), 1X GlutaMAX supplement (Thermo, 35050061), 1X MEM non-essential amino acids solution (Thermo, 11140076) and 50,000 Units/µg of Penicillin/Streptomycin (Thermo, 15140122) at 37 °C and 5% $CO_2$. Cells were seeded on a Greiner 384-Well Black Microplate with Optical-

**Fig. 5 | The PIAS4-RNF4 axis is responsible for the chromatin-associated degradation of WRN. a** HCT-116$^{WRN-Halo}$ cells were treated with the indicated siRNA oligos for 24 h, and subsequently treated with 10 μM of WRNi for 24 h before imaging. **b** Treatment of HCT-116$^{WRN-Halo}$ with 1 μM of SUMOi in the presence or absence of 10 μM WRNi for 6 h before imaging. **c, d** Quantifications of **a, b**, respectively. Graphs represent the average of $n = 3$ plates for siRNA experiments and $n = 6$ plates for the small molecule experiments. Each dot represents the average of six wells. For both **c** and **d**, error bars represent s.d. **e.** WB analysis of HCT-116 cells treated with 1 μM SUMOi in the presence or absence of 10 μM WRNi for 16 h, at which point cells were lysed and analyzed. **f** Quantification of (**e**). Graphs represent the mean value of $n = 2$ WB runs. **g** WB analysis of TUBE pulldowns of HCT-116$^{WRN-Halo}$ cells after treatment with 1 μM SUMOi in the presence or absence of 10 μM WRNi (6 h). All samples were treated with CFZ. **h** WB analysis of HCT-116 cells co-treated with WRNi in the presence or absence of E1i and/or SUMOi. **i** Dot plots of WRN diffusion coefficient via SMT after co-treatment with SUMOi and either DMSO

or WRNi. Each point represents the average diffusion coefficient across all the nuclei in an FOV. $n = 4$ plates. Lines represent sample medians. **j** Distribution of diffusive states for WRN$^{Halo}$ in HCT-116$^{WRN-Halo}$ after treatment with 1 μM of SUMOi in the presence or absence of 10 μM WRNi. The plot line represents sample means, shaded area represents s.d. **k** Dose-response of WRNi in HCT-116 cells treated with or without 10 nM SUMOi. Error bars represent s.d., $n = 3$ plates. **l** Failure to extract trapped WRN leads to persistent DNA damage. HCT-116 cells were treated with WRNi for 3 h, washed, and replenished with fresh growth medium containing 1 μM of the indicated inhibitors, then imaged. Scale bar = 10 μm. DMSO is dimethyl sulfoxide; WRNi is HRO761; E1i is TAK-243; SUMOi is ML-792; CFZ is carfilzomib. All $P$ values were calculated using a two-tailed, unpaired Student's $t$-test. MW is molecular weight. Source data are provided as a Source Data file. For all WBs, GAPDH was used as a loading control. In **g**, UB was used as a processing control; in **h**, SUMO1/2 and UB were used as processing controls.

Bottom (Greiner, 781906) using a Thermo Multidrop Combi. HCT-116 cells were seeded at 2500 cells per well and HT-29 cells were seeded at 3000 cells per well and incubated at 37 °C and 5% $CO_2$ for 24 h before compound treatment. Compounds were added to the plates using a Beckman Echo 655 acoustic liquid handler. Compound-treated plates were incubated for 24 h at 37 °C and 5% $CO_2$.

### Cell line engineering
WRN$^{Halo}$ HCT-116 and U2OS were generated by nucleofection of ribonucleoprotein (RNP) complexes (DeWitt et al., 2017) using a guide (5′ – AAAGATGAGTGAAAAAAAT–3′) targeting the N-terminus of the *WRN* gene locus and a megamer coding for the HaloTag as a donor template (5′–TATTGTATCTGTTTTGTTTTGTGATTCTAGCTCTTAT AACCTATGCTTGGACCTAGGTGTCATAACTTACTTTAAATATGTATGT TTGGTTTTCATTCATATTGACAGTACTACCTCTCAGTTTTCTTTCAGA TATTGTTTTGTATTTACCCATGAAGACATTGTTTTTTGGACTCTGCAA ATACCACATTTCAAAGATGGCAGAAATCGGTACTGGCTTTCCATTCGA CCCCCATTATGTGGAAGTCCTGGGCGAGCGCATGCACTACGTCGATG TTGGTCCGCGCGATGGCACCCCTGTGCTGTTCCTGCACGGTAACCCG ACCTCCTCCTACGTGTGGCGCAACATCATCCCGCATGTTGCACCGAC CCATCGCTGCATTGCTCCAGACCTGATCGGTATGGGCAAATCCGACA AACCAGACCTGGGTTATTTCTTCGACGACCACGTCCGCTTCATGGAT GCCTTCATCGAAGCCCTGGGTCTGGAAGAGGTCGTCCTGGTCATTCA CGACTGGGGCTCCGCTCTGGGTTTCCACTGGGCCAAGCGCAATCCAG AGCGCGTCAAAGGTATTGCATTTATGGAGTTCATCCGCCCTATCCCG ACCTGGGACGAATGGCCAGAATTTGCCCGCGAGACCTTCCAGGCCTT CCGCACCACCGACGTCGGCCGCAAGCTGATCATCGATCAGAACGTTT TTATCGAGGGTACGCTGCCGATGGGTGTCGTCCGCCCGCTGACTGAA GTCGAGATGGACCATTACCGCGAGCCGTTCCTGAATCCTGTTGACCG CGAGCCACTGTGGCGCTTCCCAAACGAGCTGCCAATCGCCGGTGAGC CAGCGAACATCGTCGCGCTGGTCGAAGAATACATGGACTGGCTGCAC CAGTCCCCTGTCCCGAAGCTGCTGTTCTGGGGCACCCCAGGCGTTCT GATCCCACCGGCCGAAGCCGCTCGCCTGGCCAAAAGCCTGCCTAACT GCAAGGCTGTGGACATCGGCCCGGGTCTGAATCTGCTGCAAGAAGAC AACCCGGACCTGATCGGCAGCGAGATCGCGCGCTGGCTGTCGACGCT CGAGATTTCCGGCGAAAACCTGTATTTTCAGAGCAGTGAAAAAAAATT GGAAACAACTGCACAGCAGCGGAAATGTCCTGAATGGATGAATGTGC AGAATAAAAGATGTGCTGTAGAAGAAAGAAAGGTATGTTGTTCATTG ACTATTCTTTTGGGTGAGAAATTTAATTTATATTTGACTGTGCAAAGA GTCAGTTGTTACTTGTAAACTTCAAGTCATTGTTTAGGTCAGAG – 3′). RNPs were assembled with Alt-R™ S.p. HiFi Cas9 Nuclease V3 (IDT, 1081060) in a 10 μl reaction of 1 μM of Cas9, 120 pmol sgRNA, and 100 pmol ssODN in Cas9 buffer (20 mM HEPES 7.5, 150 mM KCl, 10% glycerol, 1 mM TCEP). Reactions were gently mixed for 30 s and incubated for 10 min at room temperature. RNP complexes and 200k cells resuspended in 20 μl buffer SE (Lonza) were added to a nucleofection strip and the mixture pulsed with program EH-100 (Lonza 4D-Nucleofector). Cells were plated into 96-well plates for recovery with a fresh media change after 1 day. Two days after RNP electroporation,

pooled cells then were single-cell seeded into 384-well plates. After single-cell derived clones emerged, they were split into two plates, one of which was used to detect positive clones containing the desired HaloTag™ sequence through Sanger sequencing.

RNF4$^{WT}$ and RNF4$^{C132S/C135S}$ cell lines were generated using the PiggyBac transposase system. Briefly, 60 K HCT-116$^{WRN-Halo}$ cells were seeded in six-well plates and allowed to grow for 24 h. Cells were then co-transfected with 3 μg of PB-Cuo-RNF4-IRES-GFP-EF1α-CymR-T2A-Puro or PB-Cuo-RNF4$^{C132S/C135S}$-IRES-GFP-EF1α-CymR-T2A-Puro (empty vector: PBQM812A-1, System Biosciences), alongside a PiggyBac Transposase expression plasmid (PB210PA-1, System Biosciences), using Lipofectamine 2000 following manufacturer recommendations (Invitrogen, 11668027). Media was changed after 24 h, and cells were allowed to proliferate for an additional 48 h. After a total of 72 h post transfection, cells were subjected to puromycin selection (1 μg/mL) for 72 h. To induce expression of stably integrated genes, cells were treated with 1x cumate solution for at least 24 h (QM150A-1, System Biosciences).

### Antibodies and reagents
Goat anti-Rabbit IgG Alexa Fluor Plus 647 (Thermo, A32733), anti-phospho-Histone H2A.X (Ser139)-clone JBW301 (γH2A.X, Millipore, 05-636-I), anti-WRN clone EPR6392 (abcam, ab124673), Goat anti-Rabbit IgG (H + L)-HRP (Invitrogen, 31460), anti-GAPDH-HRP (CST, 31460), anti-Ubiquitin, clone P4D1 (for WB; CST, 3936), anti-Ubiquitin, clone FK2 (For IF; Millipore sigma, 04-263), anti-SUMO-2/3, clone 18H8 (CST, 4971 S), anti-phospho-CHK1 (Ser345; CST, 2341), phospho-Chk2 (Thr68; clone C13C1; CST, 2661), cleaved PARP1 (Asp214; clone D64E10; CST, 5625), anti-RNF4 (R&D Systems, AF7964), anti-MDM2 (R&D Systems, D1V2Z), anti-MIB1 (CST, 4400), anti-PIAS4 (CST, D2F12).

JF549 (Promega, GA1110), etoposide (CST, 2200), 5-FU (5-fluor-ouracil; Selleckckem, S1209), doxorubicin (CST, 5927), Halo-PROTAC3 (Promega, GA3110), HaloPROTAC-E (MedChem Express, HY-145752), MLN-4924 (CST, 85923), carfilzomib (CST, 15022), TAK-243 (MedChem Express, HY-100487), ML-792 (MedChem Express, HY-108702).

### Immunofluorescence and DNA damage quantification
The cell culture medium was removed using the Blue Cat Bio Blue®-Washer. Paraformaldehyde solution (8% PFA (Electron Microscopy Sciences, 15710-S), 1X phosphate buffered solution (PBS) (Teknova P0191)) was added to the cells via Thermo Multidrop® Combi and treated for 7 min at room temperature to fixate the cells. The paraformaldehyde solution was removed with the Blue Cat Bio BlueWasher. Triton solution (0.1% Triton X-100 (Sigma, T8787), 1X PBS) was added to the cells via Thermo Multidrop Combi and treated for 15 min at room temperature to permeabilize the cells. The triton solution was removed with the Blue Cat Bio BlueWasher. Serum solution (10% Goat Serum (Gibco, 16210), 1X PBS) was added to the cells via Thermo Multidrop Combi and treated for 15 min at room temperature to block

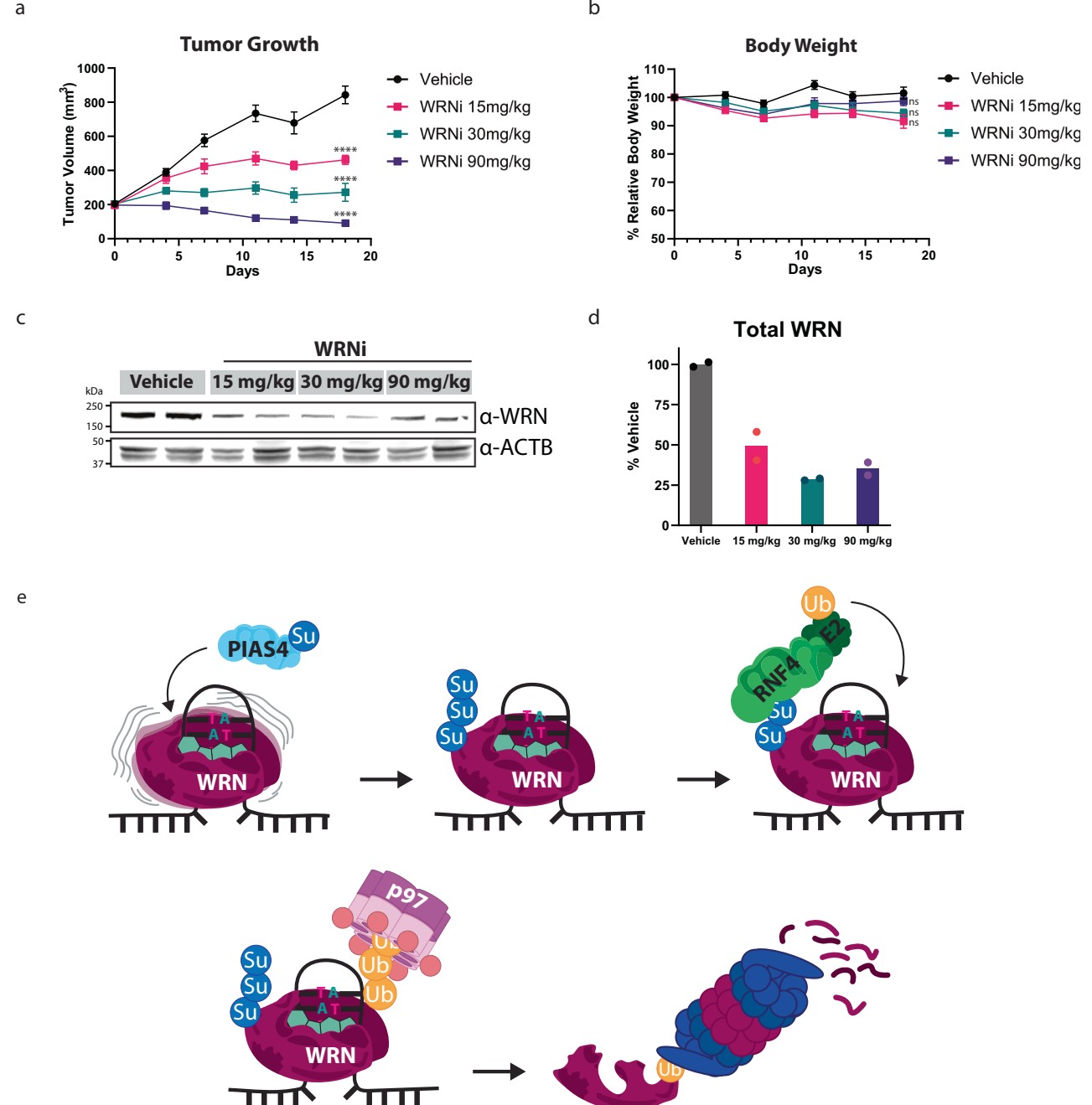

**Fig. 6 | WRN Inhibition has potent antitumor effects in an MSI-H in vivo mouse model. a** HCT-116 xenograft tumor model in nude mice shows that WRN inhibition has robust antitumor effects. Mice were dosed daily with the indicated doses for 18 days total. Error bars represent s.e.m., treatment cohorts of $n = 8$ mice. Significance was assessed via a two-way ANOVA with Dunnett's multiple comparison test; dosing groups were compared against the vehicle control at the terminal day 18 timepoint. **b** In vivo WRN inhibition is well tolerated. Body weights of mice across different treatment groups were taken throughout the study duration, showing no signs of malnourishment. Error bars represent s.e.m., treatment cohorts of $n = 8$ mice Significance was assessed via a two-way ANOVA with Dunnett's multiple comparison test; dosing groups were compared against the vehicle control at the terminal day 18 timepoint. **c** Pharmacodynamic analysis by WB of HCT-116 mice xenografts tumors after treatment with WRNi, showing the degradation of WRN. ACTB was used as a loading control. **d** Quantifications of (**c**). Lines indicate the sample median. **e** Proposed model for targeted proteasomal degradation of trapped WRN upon treatment with WRNi. Chromatin-bound WRN that is actively surveying DNA damage in MSI-H cells becomes trapped upon inhibition by WRNi. This stalled WRN is SUMOylated by the SUMO ligase PIAS4. SUMOylated WRN recruits the STUbL RNF4, leading to its ubiquitylation. Ubiquitylated WRN is extracted from chromatin by p97/VCP, leading to its degradation by the proteasome. Significance was defined as follows: ns $P > 0.01$; ****$P < 0.0001$. Source data are provided as a Source Data file.

the cells. The serum solution was removed with the Blue Cat Bio BlueWasher. Primary antibody solution (1:1000 primary antibody, 10% Goat Serum, 1X PBS) was added to the cells via a Thermo Multidrop Combi and treated for 3 hours at room temperature with shaking. The primary antibody solution was removed with the Blue Cat Bio

BlueWasher. Secondary antibody solution (secondary antibody (1:1000), Hoechst 33342 (AnaSpec, AS-83210) (1:3000), 10% Goat Serum, 1X PBS) was added to the cells via Thermo Multidrop Combi and treated for 30 min at room temperature, covered from light, with shaking. Cells were washed with PBS 3 times using the Blue Cat Bio

BlueWasher with a final dispense of PBS. Sealed plates were imaged using an ImageXpress Micro slit confocal microscope (Molecular Devices) using a 40x water immersion objective, and 6 fields of view per well. Exposure parameters were optimized to prevent pixel saturation for each channel. Images were analyzed using MetaXpress Custom Module Editor, by using a Hoechst mask to identify nuclei, then measuring the integrated Alexa Fluor 648 intensity across all nuclei in the field of view (FOV) and then averaged. Percent γH2A.X signal was calculated by using the following equation: $\%S = (T - C_{pos})/(C_{pos} - C_{neg}) \times 100$, where $\%S$ is percent γH2A.X signal and T is the measured γH2A.X fluorescence of the wells treated with test compound. The effective compound concentration leading to a 50% induction of γH2A.X signal ($EC_{50}$), and the resulting cell γH2A.X signal measured at the highest tested compound tested ($c_{max}$) was carried out by fitting a four-parameter non-linear regression using GraphPad Prism. At least three biological replicates were done per compound tested.

### In vitro western blotting
Samples were lysed in 1x NuPAGE LDS sample buffer (Invitrogen, NP0008), sonicated, and heated at 75 °C for 10 min. Samples were normalized to protein concentration and volume using a Pierce 660 reagent kit (Invitrogen, 22660). SDS-Page was performed using standard protocols (ref). WB transfers were performed using an iBlot2 nitrocellulose membrane system (Invitrogen, IB23001). Membranes were blocked with Pierce™ Protein-Free Blocking Buffer (Thermo Scientific, 37572) for 30 min at room temperature, and subsequently probed with desired antibodies.

### Chromatin fractionation assays
About $10^6$ cells were seeded on 10 cm dishes and allowed to grow for 24 h. Various drug treatments were then performed for various timepoints depending on the experiment. Following treatment duration, cells were harvested by trypsinization, centrifuged at $300 \times g$, and flash-frozen in liquid nitrogen. Cell pellets were resuspended in ice-cold CSK buffer (25 mM PIPES pH 7.5, 300 mM sucrose, 2 mM CaCl$_2$, 2 mM EGTA, 0.3% Triton X-100, 1x protease inhibitors, 1 mM DTT, 25-500 mM NaCl) for 10 min. Lysates were spun at $5000 \times g$ for 3 min at 4 °C. Supernatants were collected as the soluble fraction and denatured in LDS buffer and analyzed. Pellets containing the chromatin fraction were washed twice with CSK buffer, and subsequently solubilized in LDS buffer, followed by sonication. Fractions were then analyzed by WB.

### Cell viability assay
To seed cells, cells were trypsinized and resuspended in complete culture media to the desired concentration (HCT-116: 10,000 cells/mL; RKO: 15,000 cells/mL; SW-48, HT-29, and SW-480: 25,000 cells/mL; U2OS: 7000 cells/mL). Cell suspensions were seeded in 50 μL of complete culture media and plated onto 384-well white clear-bottom plates (Thermo Fisher Cat# 164610) using a Multidrop™ Combi liquid dispenser in the slowest setting in triplicate. After 24 h, cells were treated with compounds in a 10-pt, 3.16-step serial dilution using an Echo acoustic liquid dispenser (Beckman). DMSO was backfilled to a final concentration of 0.1%. About 10 μM etoposide ($C_{pos}$) and DMSO ($C_{neg}$) were used as reference compounds. Ninety-six hours post compound addition, plates were evacuated to decant all media. To measure viability, cellular ATP concentrations were measured by adding 20 μL of 1x CellTiterGlo 2.0 (CTG) solution (1 part PBS, 1 part CellTiterGlo 2.0 stock solution; Promega Cat# G9241) using a Multidrop™ Combi liquid dispenser and measuring the luminescence on a SpectraMax iD3 (Molecular Devices).

Percent viability was calculated by the following equation: $\%V = (T - C_{pos})/(C_{pos} - C_{neg}) \times 100$, where $\%V$ is percent viability and T is the measured luminescence of the wells treated with test compound.

The effective compound concentration leading to a 50% reduction in viability ($EC_{50}$), and the resulting cell viability measured at the highest tested compound tested ($c_{max}$) was carried out by fitting a 4-parameter non-linear regression using GraphPad Prism. At least three biological replicates were done per compound tested.

### Tandem ubiquitin-binding entities (TUBE) pull-down
HCT-116 cells were grown as described above. About $10^6$ cells were seeded on 10 cm tissue culture dishes (Thermo Fisher, 150464), and allowed to attach for 48 h. Cells were subsequently treated with compounds for 8 h ([ML-792] = 1 μM, [Carfilzomib] = 10 nM, [TAK-243] = 10 nM, [WRNi] = 10 μM), harvested by trypsinization, centrifuged ($300 \times g$), and flash-frozen in liquid nitrogen. Cell pellets were lysed in 500 μL of TUBE lysis buffer (150 mM NaCl, 2 mM MgCl$_2$, 25 mM HEPES pH 7.0, 0.03% SDS, 1 M urea, 10 μM PR-619, 1x protease inhibitor cocktail) for 10 min on ice, with frequent mixing. While cell lysis was occurring, 100 μL of slurry/reaction of TUBE magnetic beads (UM501M, LifeSensors) were washed in 1x PBS three times, and once with TUBE lysis buffer. After cell lysis, lysates were cleared ($20,000 \times g$, 10 min), inputs were taken, and lysates were subsequently added to TUBE magnetic beads. Lysate-TUBE bead mixtures were incubated at 4 °C on a nutator for 2 h. Samples were subsequently washed five times with TUBE wash buffer (150 mM NaCl, 2 mM MgCl$_2$, 25 mM HEPES pH 7.0, 0.5% Triton X-100, 1 M urea), and resuspended in 1X LDS buffer to prepare for downstream WB analysis.

### HaloTag™ protein labeling
Cells were seeded in black 384-well plates using a combidrop multi-drop dispenser and seeded for at least 24 h in complete growth media at either 45 K cells/mL or 100 K cells/mL, depending on the experiment. To label Halo-Tagged proteins, JF549 (Promega, GA1110) was added to cells using an Echo acoustic liquid dispenser to a final concentration of 25 pM. Cells were incubated at 37 °C for 2 h, and subsequently washed five times in PBS, with the final wash leaving the wells empty. After washing, cells were either fixed in 4% paraformaldehyde (Electron Microscopy Sciences, 15710), or growth media was replenished to perform subsequent compound treatments.

### WRN imaging degradation assays
Cell lines and growth conditions are identical to the above conditions. To seed cells for the assay, cells were trypsinized and resuspended in complete culture media to the desired concentration (WRN$^{Halo}$ HCT-116: 50,000 cells, WRN$^{Halo}$ U2OS: 30,000 cells/mL). Cell suspensions were seeded in 50 μL of complete culture media and onto 384-well black clear-bottom optical plastic plates (Greiner Bio-One Cat# 781097) using a Multidrop™ Combi liquid dispenser in the slowest setting in triplicate. After 24 h, cells were labeled with Halo dye as described above and subsequently treated with compounds in a 10-pt, 3.16-step serial dilution using an Echo acoustic liquid dispenser and incubated at 37 °C in a humidified 5% CO$_2$ incubator. About 10 μM Halo-PROTAC3 ($C_{pos}$; Promega, GA3110) and DMSO ($C_{neg}$) were used as reference compounds. 24 h after compound treatments, cells were fixed in 4% paraformaldehyde for 10 min, washed three times with PBS, then blocked and permeabilized in PBS containing 10% goat serum and 0.1% triton X-100 containing the DNA counterstain Hoechst for 30 min. Plates were washed three times in PBS, and sealed with thermal foil seals.

Sealed plates were imaged using an ImageXpress Micro slit confocal microscope (Molecular Devices) using a 40x water immersion objective, and 6 fields of views per well. Exposure parameters were optimized to prevent pixel saturation for each channel. Images were analyzed using MetaXpress Custom Module Editor, by using a Hoechst mask to identify nuclei, then measuring the average Halo dye intensity across all nuclei in the FOV, averaged, and then background subtracted.

Percent WRN$^{Halo}$ signal was calculated by using the following equation: %S = (T - C$_{pos}$)/(C$_{pos}$ − C$_{neg}$) × 100, where %S is the percent WRN$^{Halo}$ signal and T is the measured WRN$^{Halo}$ fluorescence of the wells treated with test compound. The effective compound concentration leading to a 50% induction of WRN$^{Halo}$ signal (EC$_{50}$), and the resulting cell WRN$^{Halo}$ signal measured at the highest tested compound tested (C$_{max}$) was carried out by fitting a four-parameter non-linear regression using GraphPad Prism. At least three biological replicates were done per compound tested.

## Phenotypic screening and imaging

An arrayed human ON-TARGETplus ubiquitome siRNA SMARTpool library (Horizon Discovery, 106205-E2-01) was resuspended to a final concentration of 20 uM in 1x siRNA Buffer (Horizon Discovery, B-002000-UB-100). siRNA oligos were transferred onto black 384 µClear plates using an Echo acoustic dispenser to yield a final oligo concentration of 20 nM. About 5 µL of opti-MEM (Gibco, 31985062) was added to resuspend transferred oligos. RNA oligo:lipid complexes were formed by adding 0.3 µL of Lipofectamine RNAiMAX (Invitrogen, 13778075) in 5 µL of opti-MEM. Complexes were incubated for 5 min before dispensing 50 µL of HCT-116 cells (100 K cells/mL) and incubating in the growth conditions described above. After 24 h, growth media was exchanged, and labeled with HaloTag™ dye as described above. WRNi, or DMSO, was added at a final concentration of 10 µM, treated for 24 h, and subsequently fixed with 4% PFA. Cells were permeabilized with 0.1% triton x-100 for 20 min to remove non-specific dye staining and imaged as described above.

## siRNA depletions

Cells were processed as described for quantifying WRN levels after siRNA knock-down. For WB analysis, HCT-116 (90 K cells/well) or U2OS (60 K cells/well) cells were seeded in six-well TC treated plates (Corning, 3516). siRNA depletions were performed following the recommended protocol for Lipofectamine RNAiMax. Samples were harvested in 1x LDS sample buffer and prepared for WB analysis as described.

## WRN in-situ trapping assay

WRN$^{Halo}$ Cells (HCT-116 or U2OS) were seeded in black 384-well plates using a combidrop multidrop dispenser and seeded for 48 h in complete media at 100 K cells/mL (HCT-116) or 30 K cells/mL (U2OS). Cells were labeled with HaloTag™ Dye and subsequently treated with desired compounds as described above for 8 h ([ML-792] = 1 µM, [Carfilzomib] = 10 nM, [TAK-243] = 10 nM, [WRNi] = 10 µM). Following compound treatments, samples were decanted to remove all media and treated with ice-cold CSK buffer (25 mM PIPES pH 7.0, 300 mM NaCl, 2 mM MgCl$_2$, 0.3 % Triton X-100, 200 mM sucrose) for 2 min, on ice. Without removing buffer in wells, cells were fixed in 4% PFA supplemented with Hoechst for 15 min and washed with 1x PBS five times using an AquaMax Plate washer (Molecular Devices). Samples were then imaged as described above. Quantifications were done by measuring the total intensity of the Halo dye signal and using the Hoechst channel as a nuclear mask.

## Protein purification

Human WRN residues 480–1251 were cloned into pFastBac vector containing an 8xHis N-terminal tag and expressed in Sf9 insect cells. Harvested cells were resuspended in lysis buffer (50 mM HEPES, pH 7.5, 500 mM NaCl, 25 mM imidazole, 1 mM TCEP, 5 U/mL benzonase (Millipore Sigma), EDTA-free cOmplete protease inhibitor cocktail tablet (Roche)) and lysed by addition of Insect PopCulture reagent (Millipore Sigma). Lysate was loaded onto a HiTrap TALON Crude column (Cytiva) and protein was eluted using 500 mM Imidazole. Fractions containing WRN protein were collected, pooled, and then diluted in a buffer to drop the NaCl concentration to ~100 mM. The

pooled and diluted sample was then loaded onto a heparin column (Cytiva) and eluted using a step-wise gradient from 200 mM to 1000 mM NaCl. Fractions containing purified WRN protein were pooled, and buffer exchanged into 50 mM HEPES, 150 mM NaCl, 10% glycerol, 1 mM TCEP using a HiPrep desalting column (Cytiva).

Human BLM residues 636-1298 were cloned into pFastBac vector containing an 8xHis N-terminal tag and expressed in Sf9 insect cells. Harvested cells were resuspended in lysis buffer (50 mM HEPES, pH 8, 200 mM NaCl, 0.5 mM TCEP, 5 U/mL benzonase (Millipore Sigma), EDTA-free cOmplete protease inhibitor cocktail tablet (Roche)) and lysed by addition of Insect PopCulture reagent (Millipore Sigma). Lysate was loaded onto a Ni-NTA column (Thermo Fisher) and protein was eluted using 300 mM Imidazole. Fractions containing BLM protein were collected, pooled, and diluted in a buffer to drop the NaCl concentration to ~50 mM. The pooled and diluted sample was then loaded on a HiTrap Heparin column (Cytiva) and eluted with a linear gradient from 50 mM to 1 M NaCl. The pure fractions were pooled, concentrated, and purified by size exclusion chromatography using S200 Increase (Cytiva) in 50 mM HEPES pH 8, 200 mM NaCl, 5% glycerol, 0.5 mM TCEP.

## Helicase unwinding and ATPase assay

The helicase unwinding and ATPase were carried out in a multiplexed fashion based on previously published BLM assay[53]. Single-stranded DNA was purchased from IDT:

A: 5'-Cy3-GAACGAACACATCGGGTACGTTTTTTTTTTTTTTTTTTTTTTTTTTTTTTTTT

B: 5'-TTTTTTTTTTTTTTTTTTTTTTTTTTTTTTTTTTCGTACCCGATGTGTTCGTTC-IowaBlackFQ-3'

B-dark: 5'-CGTACCCGATGTGTTCGTTCY-3'.

Strands A and B were annealed in TE + 50 mM NaCl in a slowly-cooling thermocycler.

First, compounds in DMSO were dispensed to a 384-well white ProxiPlate (Perkin Elmer) using an Echo acoustic liquid handler. WRN or BLM Protein was diluted into assay buffer and 2.5 µL was dispensed into each well. Protein and compound were then incubated at room temperature for 15 min. Following pre-incubation, 2.5 µL of DNA and ATP in assay buffer was dispensed into each well to initiate the reaction and incubated for 20 min at room temperature. At the 20-min timepoint, Cy3 fluorescence was read on an Envision plate reader to measure unwinding activity. ATP hydrolysis was then measured using ADP-Glo kit (Promega) by adding 5 uL ADP-Glo reagent for 40 min followed by 10 uL Kinase Detection Reagent for 1 h. Luminescence was measured using an Envision plate reader. Data were normalized to DMSO (100% activity). The final reaction conditions are: 1 mM ATP, 15 nM dsDNA substrate, 1.5 uM B-dark in reaction buffer composed of 50 mM Tris-HCl pH 8.0, 50 mM NaCl, 2 mM MgCl$_2$, 0.01% Tween-20, 2.5 µg/mL poly(dI-dC), 1 mM DTT, 1% DMSO, and 12.5 nM WRN or 2.5 nM BLM.

## Cellular SMT sample preparation

For SMT experiments, WRN Halo-tagged HCT-116 cells were seeded in FluoroBrite DMEM (Thermo Fisher, cat. no. 1896701) supplemented with 10% FBS (Corning), 1% Penicillin/Streptomycin (Gibco, 1510-122), and 1% GlutaMAX (Gibco, 35050-061) on plasma-coated 384-well glass-bottom plates (Cellvis, P384-1.5H-N) at $1.5 \times 10^4$ cells per well. WRN Halo-tagged U2OS cells were seeded in GlutaMAX-supplemented DMEM (Gibco, 10566-016) with 10% FBS and 1% Penicillin/Streptomycin at $6 \times 10^3$ cells per well. Prior to treatment and imaging, HCT-116 cells were incubated at 37 °C and 5% CO$_2$ for ~48 h to allow for cell adherence to the plates, while U2OS cells were incubated under the same conditions for 24 h. For all SMT experiments, cells were treated with 10–40 pM of JF549-HTL (synthesized in-house) and 200 nM Hoechst 33342 (Thermo Fisher, cat. No. 62249) for 1 h in complete medium at 37 °C and 5% CO$_2$. Cells were then washed three times in PBS and twice in FluoroBrite DMEM supplemented with 2%

FBS, 1% Penicillin/Streptomycin, and 1% GlutaMAX. All compounds were prepared on Echo Qualified 384-Well Low Dead Volume Source Microplates (Labcyte, cat. No. LP-0200) in DMSO and administered onto cells at a final 1:500 dilution in cell culture medium. Unless otherwise specified, cells were incubated with compounds for 4 h at 37 °C prior to image acquisition. When possible, well-replicate conditions were randomized across each plate. For all experiments, control conditions included vehicle (DMSO) treatment and wells lacking JF549-HTL to assess possible effects from the detection of non-dye signal.

## Cellular SMT image acquisition

Unless otherwise stated, all image acquisition for SMT was performed on a customized Nikon Eclipse Ti2-E inverted fluorescence microscope with a motorized stage. The microscope system was outfitted with a stage top environmental chamber with temperature and $CO_2$ control (OKO labs), Nikon objective water dispenser, an Oblique Line Scanning (OLS) illumination module (Driouchi et al., 2023) with laser launch containing 405, 560, and 642 nm lasers, three-band emission filter set (ET 445/58 m, FF01-585/40-25, FF01-676/37-25, Chroma), motorized filter wheel (Lambda 10-B; Sutter Instruments), and a high-speed sCMOS camera equipped with light-sheet mode capability (ORCA-Fusion BT, Hamamatsu). Images were acquired with a 60×1.27 NA water immersion objective (CFI SR Plan Apo IR 60XC WI, Nikon, Japan). The environmental chamber was maintained at 37 °C, 95% humidity, and 5% $CO_2$. For each field of view, 150 SMT frames were collected at a frame rate of 100 Hz with a 407-microsecond stroboscopic laser pulse, and 1 frame in the Hoechst channel was subsequently collected for downstream registration of trajectories to nuclei. Each frame captured an FOV that was 1728 × 2304 pixels (187.14 × 249.52 microns) in size. Automated microscope control and image acquisition was performed using customized scripts in MicroManager.

For all experiments, conditions were tested by acquiring JF549 movies and Hoechst images at 4 FOVs per well, a minimum of two well replicates per plate, and a minimum of two plate replicates. Unless otherwise stated, reported averages for each condition are the mean value of all FOVs collected. For time course experiments, reported averages at each timepoint are the mean value of all FOVs collected across six consecutive wells per plate replicate.

## Cellular SMT image processing

Image acquisition yielded one JF549 movie and one Hoechst movie per FOV. The JF549 movie was used to track the motion of individual JF549 molecules, while the Hoechst movie was used for nuclear segmentation.

For tracking, we used a pipeline that operates in three sequential steps:

1. Detection. Fluorescent emitters are detected using a generalized log-likelihood ratio detector[54].
2. Subpixel localization. The position of each detected emitter is estimated using a Levenberg–Marquardt fitting routine to a 2D integrated Gaussian spot model[55–58]. This routine begins from an initial guess provided by the radial symmetry method[59].
3. Linking. Localized emitters are linked through time into trajectories using a probabilistic modification of Sbalzerini's hill-climbing algorithm[60,61].

Here we describe these three steps in more detail. During the detection step, we consider whether each 11 × 11-pixel tile in an image contains a 2D Gaussian spot with width of $\sigma_w = 1.5$ pixels, under an assumption of a Gaussian-distributed noise background. A useful solution to this problem is provided by generalized log-likelihood ratio testing[54]. In this framework, we consider the relative likelihoods of two competing hypotheses for an observed tile **X**:

A. $X = b_0 + \sigma_0 N$, where the right-hand side is white Gaussian noise with mean $b_0$ and variance $\sigma_0^2$.

B. $X = b_1 + \sigma_1 N + IG$, where $b_1 + \sigma_1 N$ is white Gaussian noise with mean $b_1$ and variance $\sigma_1^2$, $I$ is an unknown intensity, and **G** is a 2D Gaussian with width $\sigma_w = 1.5$ pixels, mean 0, and energy 1.

To determine whether the tile contains a spot, we compute the log-likelihood ratio LLR = log p(**X** | hypothesis B) − log p(**X** | hypothesis A), where the probabilities are evaluated at the max likelihood estimates for the parameters of the respective hypothesis. It can be shown[54] that the LLR can be written

$$LLR = -\frac{w^2}{2}\log\left[1 - \frac{\left(\sum_{i,j}^{w} X_{ij} G_{ij}\right)^2}{\sum_{i,j}^{w} X_{ij}^2 - \frac{1}{w^2}\left(\sum_{i,j}^{w} X_{ij}\right)^2}\right] \quad (1)$$

Here $w$ is the tile size (11 pixels) and the sums run over all $w^2$ pixels in the tile. In practice, the three sums are computed simultaneously for all tiles in the image using frequency-domain convolution. We detect a spot in any tile for which LLR ≥14; this threshold was chosen empirically to balance detection recall and false detection rate.

In the subpixel localization step, we estimate the location of each emitter more precisely by fitting the observed spot to a 2D integrated Gaussian model. This model is identical to that used by Smith et al., 2010 and can be expressed:

$$\Delta x_i = \frac{1}{2}\left[\text{erf}\left(\frac{x_i - x_0 + 1/2}{\sqrt{2\sigma_w^2}}\right) - \text{erf}\left(\frac{x_i - x_0 - 1/2}{\sqrt{2\sigma_w^2}}\right)\right] \quad (2)$$

$$\Delta y_j = \frac{1}{2}\left[\text{erf}\left(\frac{y_j - y_0 + 1/2}{\sqrt{2\sigma_w^2}}\right) - \text{erf}\left(\frac{y_j - y_0 - 1/2}{\sqrt{2\sigma_w^2}}\right)\right] \quad (3)$$

$$f_{i,j} = I\Delta x_i \Delta y_j + b \quad (4)$$

where $f_{i,j}$ is the spot model evaluated at pixel $(x_i, y_j)$ in the tile. The free parameters are $x_0$ (x position), $y_0$ (y position), $I$ (intensity), and $b$ (offset); the spot width $\sigma_w$ is constrained to 1.5 pixels. We fit this model to the observed 11 × 11-pixel tile around a detected emitter using a Levenberg–Marquardt fitting routine in which the parameter vector $\theta = (x_0, y_0, I, b)$ is updated according to $\theta^{(t+1)} = \theta^{(t)} - \gamma \left(J^T J + \alpha I\right)^{-1} J^T \left(X - f\left(\theta^t\right)\right)$. Here, $\theta^{(t)}$ is the parameter vector at iteration $t$, $J$ is the Jacobian corresponding to the spot model above, $f\left(\theta^{(t)}\right)$ is the spot model evaluated at parameters $\theta^{(t)}$, $X$ is the vector of gray values corresponding to the observed spot, and $\gamma = 0.3$ and $\alpha = 10^{-4}$ are fitting hyperparameters. We run at most eight iterations, calling convergence if $x_0$ and $y_0$ change by less than $10^{-4}$ at any given iteration. The Levenberg–Marquardt routine is initialized from a guess in which $x_0^{(0)}$ and $y_0^{(0)}$ are estimated by the radial symmetry method[59].

In the linking step, we first construct a graph of possible identity relations (i.e., links) between detected emitters, then select a subset of links from this graph to include in trajectories. The graph of potential links is constructed by making a link for every two detections separated by up to 1.25 μm in space and up to three frames in time. The probability of each link in this graph is estimated with respect to a Brownian motion model using a variational optimization algorithm that jointly infers marginal link probabilities and the diffusion coefficients of each spot. Once the marginal link probabilities are inferred, we use a hill-climbing algorithm[61] to find the subgraph of links that maximizes the sum of marginal link log probabilities under the constraint that no two links may begin or end at the same detection. Trajectories are then called as connected components in this subgraph.

For nuclear segmentation, all frames of the Hoechst movie were averaged to generate a mean projection. This mean projection was then segmented with a UNET-based convolutional neural network trained on human-labeled nuclei[62]. Each spot was then assigned to at most one nucleus using its subpixel coordinates. To recover dynamical information from trajectories, we used state arrays[63], a Bayesian inference approach, with the RBME likelihood function and a grid of 100 diffusion coefficients from 0.01 to 100.0 $\mu m^2$/sec and 31 localization error magnitudes from 0.02 $\mu m$ to 0.08 $\mu m$ (1D RMSD). After inference, localization error was marginalized out to yield a one-dimensional distribution over the diffusion coefficient for each FOV.

### In vivo mice xenograft efficacy and pharmacodynamic (PD) studies

All mice studies were performed at the Mispro Vivarium facility in New York, NY, and were conducted according to the guidelines of the Mispro Institutional Animal Care and Use Committee and Eikon Therapeutics protocol. Specifically, mice are housed in four animals per cage, provided nestlets and disposable huts, and allowed free access to food and water under standard environmental vivarium conditions (temperature of 72° ± °F with a 12 h light cycle). The IACUC protocol for these studies is EIK-01.

For efficacy studies, $2.0 \times 10^6$ HCT-116 cells were implanted into the right flank of 8-week-old female Crl:NU(NCr)-Foxn1nu outbred mice (Charles River, strain code: 490). Body weight was taken, and tumors were measured with calipers twice per week for the duration of each study. Tumor volume was calculated using the formula TV = 0.5 × length × width$^2$. After HCT-116 tumors grew to an average volume of 150–200 mm$^3$ (7 days post-implantation), mice were randomized into treatment groups ($n = 8$ mice/group) and received either WRNi or vehicle control daily via oral gavage. At the end of the study, tumors were collected 4 h post treatment and snap-frozen in liquid nitrogen. Tumor burden was monitored closely to ensure that no tumor exceeded 2 cm (20 mm) growth in diameter in any one direction. One mouse in this study exceeded this limit, with a tumor reaching 2.1 cm diameter in one direction resulting in immediate euthanasia.

For PD studies, Mice were treated with either WRNi or vehicle, then 4 h post treatment were euthanized via CO2. Tumors were immediately excised, collected and snap-frozen in liquid nitrogen. Samples were placed on dry ice, then stored at −80 °C until they were processed for WB analysis.

### In vivo Western blot analysis

Frozen xenograft samples were lysed using an SDS lysis buffer (1% SDS, 50 mM Tris pH 8.0, 10 mM EDTA, 100 mM NaCl, 1x protease and phosphatase inhibitor cocktail), and quantified using Pierce BCA Protein Assay Kit (Thermo Scientific, 23227). Then, 10 ug per sample was separated by SDS-PAGE and transferred to the Nitrocellulose membrane using an iBlot2 transfer system. The blots were imaged using KwikQuant Digital-ECL (Kindle Biosciences) after being incubated with primary antibodies Anti-WRN (Abcam, ab124673, 1:1000) and Anti-beta Actin (Abcam, ab8226, 1:1:000) and the appropriate anti-species HRP-conjugated secondary antibodies (Kindle Biosciences, R1006, R1005).

### Synthesis of HRO761

The WRN inhibitor HRO761 was synthesized following an established synthesis route for Compound 42, as described in WO2022/249060. (Supplementary Figs. 11–13)[13].

### Reporting summary

Further information on research design is available in the Nature Portfolio Reporting Summary linked to this article.

### Data availability

All non-SMT data pertinent to the findings presented in this article can be found within the article, Supplementary Figs., and source data. Processed SMT data can be found within the source data file. However, there is no appropriate database in existence at present in order to deposit our raw Single-Molecule Tracking (SMT) data, and similarly, the computing power necessary to process the results is not accessible to most. Parties interested in accessing the data contained within the manuscript should contact the corresponding authors to determine which non-proprietary data are needed and in what format they will be provided through a Data Use Agreement with Eikon Therapeutics. Other materials will be made available through a Materials Transfer Agreement with Eikon Therapeutics by contacting the corresponding author. Source data are provided with this paper.

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

## Acknowledgements

The authors extend their deepest gratitude to all the employees and consultants of Eikon, past and present, especially, Pratir Doshi, Nick Vaquera, Jeff Dove, Tiffany Cheng, Roma Moore, Bruno da Rocha Azevedo, Madhu Mena, Emily Kirkeby, Adi Hanuka, Brynmor Davis, Puneet Kumar, Dave Piotrowski, Melissa Dumble, Connie Wong, Geeta Sharma, and Alex Therien. Their tireless work enabled the experiments described here. We thank Roger Perlmutter and Robert Tjian for helpful discussions and critical feedback on the direction of our investigation and on the resulting manuscript. Eikon Therapeutics provided all funding.

## Author contributions

F.R.P. conceived the project; F.R.P., D.N., L.S., C.B., S.B., D.A., and J.H. designed the experiments; Y.T. and E.G. generated the Halo-Tagged cell lines; D.M., D.N., A.H., R.B., A.T., W.T. and L.S. performed SMT imaging and analysis; M.H., H.M. and J.O. helped F.R.P with imaging experiments and analysis; M.L. coordinated the synthesis of WRNi; R.G.M., J.F., K.C., and Z.Z. purified protein, and developed and executed the biochemical activity assays; K.V.B., D.M., and M.K. performed the in vivo studies; F.R.P., C.B., S.B., D.A., R.M.M., and D.N. wrote the manuscript. All the authors discussed the results and methods and contributed to the manuscript.

## Competing interests

The authors are employees and/or shareholders of Eikon Therapeutics, Inc. Furthermore, a patent application related to the subject matter described in the manuscript has been filed by Eikon Therapeutics, Inc. F.R.P. and S.B. are listed as co-inventors on US provisional patent application 63/599,976. The authors declare no other competing interests.
