## [Peer Review File · Nature Communications]

WRN Inhibition Leads to its Chromatin-Associated Degradation via the PIAS4-RNF4-p97/VCP axisREVIEWER COMMENTS

Reviewer #1 (Remarks to the Author):

Pérez et al. identify a novel molecular pathway that regulates WRN activity upon WRN inhibition. Utilizing a single molecule tracking platform, they gain insights into dynamic localization of WRN within live cancer cells. Mechanistically, the authors come up with that small molecule inhibition of WRN leads to its trapping onto chromatin, leading to its SUMOylation and subsequent ubiquitylation by RNF4 which ultimately leads to proteasomal degradation.

The findings are original and interesting. However, for comparison to contemporary literature of Nature Communications, additional experiments are required.

Major concerns:

1. Biochemistry test should be provided. SUMOylation and ubiquitylation of WRN need more details.

Small molecule inhibition of WRN leads to its SUMOylation by PIAS4 and subsequent ubiquitylation by RNF4. It is necessary to provide more evidences about this process, for instance, ascertaining the interactions by endogenous or exogenous Co-IP assays; Finding the amino acid residues of both SUMOylation and ubiquitylation by mass spectrum; Determining types of ubiquitination, K48/K63/etc. Mutating the identified amino acid residues of WRN and then examining the SUMOylation and ubiquitylation state.

2. Classic experiments should to be designed as a positive control of the single molecule tracking (SMT) platform.

By utilizing single molecule tracking, the author finds that upon WRN inhibition, WRN becomes trapped on chromatin. It is necessary to reveal this process by classic chromatin-bound and nuclear soluble fractions of WRN, instead of using a single method. In extended data Fig5, chromatin-bound assay and nuclear soluble fractions of WRN upon WRNi, WRNi+CFZ, WRNi+p97i, WRNi+E1i, WRNi+SUMOi should be provided. Besides, the authors are supposed to analyze their data across traditional means.

3. For the functional part, although it shows the synergistic shift in the EC50 of WRNi upon co-inhibition of the SUMOylation cascade or depletion of RNF4, it is doubtful that it has gigantic therapeutic intervention values. Inhibition of SUMOylation or E3 ligase may be detrimental and cause side effects since it has abundant potential targets. Co-inhibition molecules should be precise as much as possible. The better strategy is to figure out the molecular docking mode of WRN with PIAS4 and screen specific drugs which can block their interactions.

4. Related to point 3, animal test should be performed.

It will be nice to see cell viability under the treatment of WRNi combined with new drug screened. More importantly, it is necessary for the survival state of animals with MSI-H xenograft tumor under the treatment of WRNi combined with inhibition of the SUMOylation, depletion of RNF4, or drugs screened.

5. How does the trapped WRN recruit SUMO ligase PIAS4?

6. Fig5a, siPIAS4 should be validated by western blot.

Minor points:

1. Subtitles should be provided in the 'MAIN' part.

2. As Fig3c shows the degradation of WRN-Halo under 10 μ M WRNi for 16 h, how are the earlier images every few hours before it comes to 16h.

3. In Fig5a,5b, why is WRN-Halo partially located in cytoplasm?

Reviewer #2 (Remarks to the Author):

WRN Inhibition Leads to its Chromatin-Associated Degradation Via the PIAS4-RNF4-p97/VCP Axis

Rodríguez Pérez et al.

Authors present work aimed at understanding the mechanism of action of a novel WRN helicase inhibitor. In brief the approach used involved cell biology and biochemical methods to characterise the effect of a small molecule WRNi on its target, WRN. In particular, the authors used advanced high-throughput imaging and analysis to extract information about the properties and behaviour of WRN under inhibition.

Major conclusions

The authors demonstrate that:

- WRNi is synthetic lethal in MSI-H cells, but not in MSS ones.
- WRNi leads to “trapping” of WRN on chromatin.
- Trapped WRN is destabilized and quickly degraded through an Ub-mediated pathway.
- WRN degradation follows the PIAS4-RNF4-p97 paradigm for the removal of chromatin trapped proteins.

Is the question addressed important?

MSI-H is a relatively common biomarker-defined cancer subgroup. The potential to target MSI-H cancers with WRNi is of a significant translational interest. As this work describes a novel WRNi it merits a significant interest.

Are the conclusions novel and will they influence thinking in the field?

The conclusions of the paper are not entirely novel as it was previously demonstrated that another small molecule WRNi inhibitor (NSC617145) can trap WRN on DNA. A number of other papers have shown WRN is synthetic lethal in MSI-H cells, and yet another set of papers have shown how chromatin-trapped proteins are extracted by the PIAS4-RNF4-p97 cascade. Hence, the authors have put all of these into a consistent model showing that the novel WRNi combines the known observation of synthetic lethality, trapping and chromatin extraction.

The authors have not provided any head-to-head comparisons with previously published inhibitors; neither have they assessed the in vivo potential of the new small molecule inhibitor, which would have been of a significant interest. Collectively, this undermines the potential impact of the work.

Quality of the data provided

The data provided in the manuscript is generally of a good quality. It is well-illustrated, analysed and communicated efficiently. Where it suffers is at certain points of clarity (detailed below) and consistency – showing the same set of experimental observations in different settings.

The data is communicated with sufficient detail in order to understand the main points of the experiment but some of the key underlying data (e.g. the siRNA screen data or the numerical data extracted from the image analysis) is not included and should be. The statistical analysis also lacks consistency of presentation, for example on bar plots sometimes individual values are included, but sometimes they are missing; the information on the number of reps and error bars could be improved.

Major points

1. In general, the novel WRNi should be characterised in a larger panel of cell lines, so that some comparison to the genetic dependency upon WRN in MSI-H tumor cells lines can be made. The authors have largely used HCT-116 cells, and to a degree U2-OS. They should demonstrate the effect of the drug on viability in a larger set of MSI-H cells (OVK18, OVK12, SW48, SNU1 etc.). Could the authors also compare this small molecule to NSC617145 in a head-to-head manner?
2. To address the specificity of the drug, besides the in vitro characterisation they provide, it would be important to test the drug in WRN KO models of which there are a few (MCF7, U2OS, HAP1). Obviously, these are not MSI-H cells (or else the KO will be synthetic lethal), but they still can provide information about this drug's specificity. For example WRN genetic inhibition causes PARP

inhibitor sensitivity; does the small molecule inhibitor exacerbate PARPi sensitivity in WRN k/o cells or is it epistatic (the critical control here would be to show PARPi sensitivity with the inhibitor in WRN WT cells).

3. In vivo assessment of the effect of the new WRNi on MSI-H vs MSS cells is of a great importance.

4. Another general point about the data in all the biochemical figures is about the annotation of which cell lines are used. Sometimes these are HCT116 and sometimes their Halo-WRN HCT-116. This is important as it seems that the authors have tagged only one of the alleles (how many are they in HCT-116?), which may interfere with the interpretations of their single molecule results. They could at least discuss this point.

5. More importantly, from Extended Data Figure 2d it seems that WRNi degrades the Halo-WRN less efficiently than the untagged protein. Is this the case and how does this affect the rest of the results? On Figure 3a we see only a single band on the WB yet these are the HCT-116 Halo-WRN cells. Can the authors explain this discrepancy? Also they should provide full WB images for such experiments besides the cropped bands.

6. The authors should demonstrate WRN trapping by conventional methods before basing all their conclusions on image-based analysis of a tagged protein. The standard assay for trapping is chromatin fractionation followed by WB. This data should be included in a revised Figure 2. Similarly, key statements about the removal of trapped WRN should be supported with biochemical data as well (chromatin fractionation), for example that data in Figure 4h-k, Figure 5i-j.

7. The data on RNF4, PIAS4, p97 is convincing, but it could be further strengthened by the use of orthogonal approaches e.g. the use of dominant negative constructs, which do exist for RNF4 and p97. In addition, the data in figure 4 and 5 should be made consistent for each of the steps of the pathway. At the moment, there are mismatching pieces of information e.g. pH2AX for some, diffusion data for others.

8. Of note, the data presented in Extended Data Figure 6e indicates that PIAS4 depletion has a very strong viability effect, which makes the interpretation of results from such cells unreliable.

Minor points

Extended Figure 3C – have different loading controls and normalisations for HCT-116 and U2-OS. This needs to be made consistent.

Extended data 6f-h – please calculate Bliss coefficient for synergy/supra additivity.

Reviewer #3 (Remarks to the Author):

The manuscript written by Pérez et al. studied the effect of WRN inhibition on WRN's dynamic localization within the nuclei of live cancer cells using single-molecule tracking method. The results show that WRN inhibition traps the helicase on chromatin, requiring its degradation via the p97/VCP pathway, a process unique to MSI-H cells. The research identifies the PIAS4-RNF4 axis as crucial for WRN degradation and shows that simultaneous inhibition of WRN and SUMOylation is particularly toxic to MSI-H cells, suggesting a new regulatory mechanism for WRN that could lead to targeted cancer treatments.

I was asked to evaluate the single-molecule tracking experiments in this manuscript. Therefore, I will only comment on the SMT data and methodology. Overall, the SMT method used in this manuscript and the presented SMT data are sound. However, I have several concerns listed below:

1. The light sheet-based SMT platform needs to be more detailed described in this manuscript. Only mention "Oblique Line Scanning (OLS) illumination module" is not enough.
2. How many data points were used to calculate the diffusion coefficient in Figure 2b? There are many red lines in Figure 2b are straight line segments. Does this line only have two data points? What is the precision of the calculated diffusion coefficient in Figure 2b?

3. To validate the proposed SMT method, a control experiment needs to be performed. I suggest authors track 60 nm fluorescent beads in a water solution and calculate the diffusion coefficient of these beads to demonstrate the $6 \mu\text{m}^2/\text{s}$ tracking speed.
4. It is impressive the authors can extract the molecule's moving trajectory from blurry images (e.g., Figure 1a). It is necessary to provide the raw images data and present the outcome of each data process step. The data processing parameters also need to be provided.
5. What is the thickness of the light sheet? For a fast-moving molecule, for example, $6 \mu\text{m}^2/\text{s}$, it is very easy to diffuse out the excitation focal plane. How do you overcome this problem?

May 5th, 2024

Dear Reviewers,

We have now completed the additional experiments requested by the reviewers of our WRN manuscript. This includes:

- Additional data around our WRN single molecule tracking (SMT) assay
- Chromatin precipitation data that conclusively validates our SMT and imaging observations that WRN inhibition results in reduced WRN diffusion due to the protein being trapped on DNA
- Chromatin precipitation data that further strengthens the link between trapped WRN and p97/VCP.
- Additional validation of the WRN ubiquitylation observation using the expression of wild type and mutant forms of RNF4 in MSI-H cells
- *In vivo* efficacy data showing that the biological pathways we describe upon WRN inhibition translate to reduced tumor growth in an MSI-H xenograft model.

We thank you for your insightful comments. We believe that this data substantially strengthens the manuscript and our overall contribution to the understanding of WRN regulation in MSI-H cancer cells. As requested, we specifically address each reviewer's points below. We are excited to re-submit the updated manuscript for consideration of publishing in Nature Communications.

Cheers,

Steve Basham and Fernadno Rodriquez Perez

REVIEWER COMMENTS

Reviewer #1 (Remarks to the Author):

Pérez et al. identify a novel molecular pathway that regulates WRN activity upon WRN inhibition. Utilizing a single molecule tracking platform, they gain insights into dynamic localization of WRN within live cancer cells. Mechanistically, the authors come up with that small molecule inhibition of WRN leads to its trapping onto chromatin, leading to its SUMOylation and subsequent ubiquitylation by RNF4 which ultimately leads to proteasomal degradation. The findings are original and interesting. However, for comparison to contemporary literature of Nature Communications, additional experiments are required.

Major concerns:

1. Biochemistry test should be provided. SUMOylation and ubiquitylation of WRN need more details.

Small molecule inhibition of WRN leads to its SUMOylation by PIAS4 and subsequent ubiquitylation by RNF4. It is necessary to provide more evidences about this process, for instance, ascertaining the interactions by endogenous or exogenous Co-IP assays; Finding the amino acid residues of both SUMOylation and ubiquitylation by mass spectrum; Determining types of ubiquitination, K48/K63/etc. Mutating the identified amino acid residues of WRN and then examining the SUMOylation and ubiquitylation state.

Updated manuscript, See Fig. 4g-j and Extended Data Fig. 6 g: We decided that it would be more biologically meaningful to address this request in live cells compared to biochemical reconstitution. To achieve this, we utilized an inducible cellular expression system in combination with siRNA knock-down. We first show that in WRN inhibited cells, knock-down of RNF4 prevents WRN ubiquitylation and that this phenotype can be rescued by reintroducing wild-type RNF4 that is resistant to the siRNAs. We then show that overexpression of a catalytically dead version of RNF4 is unable to rescue ubiquitylation of inhibited WRN. We then demonstrate a direct ubiquitylation of WRN by RNF4 after inhibitor treatment using a TUBE pull-down approach. These data provide direct and conclusive evidence that RNF4 is indeed the ligase responsible for ubiquitylating inhibited WRN. While we appreciate the reviewer's suggestion for a mass spectrometry experiment, we believe that this additional data addresses those concerns.

2. Classic experiments should to be designed as a positive control of the single molecule tracking (SMT) platform.

By utilizing single molecule tracking, the author finds that upon WRN inhibition, WRN becomes trapped on chromatin. It is necessary to reveal this process by classic chromatin-bound and nuclear soluble fractions of WRN, instead of using a single method. In extended data Fig5, chromatin-bound assay and nuclear soluble fractions of WRN upon WRNi, WRNi+CFZ, WRNi+p97i, WRNi+E1i, WRNi+SUMOi should be provided. Besides, the authors are supposed to analyze their data across traditional means.

Updated manuscript, See Fig. 3g-i, Extended Data Fig. 4g-i, and Extended Data Fig. 5f: We thank the reviewer for suggesting these critical experiments as we believe

these experiments validate our platform and strengthen the biological conclusions of this paper. To provide additional validation of our SMT observations, we performed cell fractionation and chromatin precipitation experiments. We show that WRN protein remains bound to chromatin upon treatment with the WRN inhibitor in a dose-dependent manner. We also show that inhibition of VCP extends the timing where WRN can be detected by chromatin precipitation which is consistent with VCP's role in extracting "trapped" WRN from DNA. Lastly, as expected, we show that the binding of WRN to chromatin after inhibitor treatment is specific to MSI-H cancer cells as we don't observe this association after WRNi treatment in the MSS cell line U2OS.

3. For the functional part, although it shows the synergistic shift in the EC50 of WRNi upon co-inhibition of the SUMOylation cascade or depletion of RNF4, it is doubtful that it has gigantic therapeutic intervention values. Inhibition of SUMOylation or E3 ligase may be detrimental and cause side effects since it has abundant potential targets. Co-inhibition molecules should be precise as much as possible. The better strategy is to figure out the molecular docking mode of WRN with PIAS4 and screen specific drugs which can block their interactions.

Updated manuscript: While we appreciate the nature of this comment, defining the molecular docking mode of WRN with PIAS4 is outside the scope of this manuscript.

4. Related to point 3, animal test should be performed.

It will be nice to see cell viability under the treatment of WRNi combined with new drug screened. More importantly, it is necessary for the survival state of animals with MSI-H xenograft tumor under the treatment of WRNi combined with inhibition of the SUMOylation, depletion of RNF4, or drugs screened.

Updated manuscript, See Fig. 6a-d: There was universal interest among all three reviewers for in vivo data that supports our in vitro findings. To this end, we established an MSI-H xenograft model in mice using the human colorectal cancer cell line HCT-116. We then demonstrate that daily oral dosing of HR0761 results in a dose-dependent decrease in tumor growth in these animals. Western blot analysis of tumor samples from this study shows a dose-dependent decrease in WRN levels which is consistent with our in vitro findings that inhibited WRN is actively targeted for degradation.

5. How does the trapped WRN recruit SUMO ligase PIAS4?

Updated manuscript: This is an interesting question and is something that the team is currently investigating. We do not have new data that addresses this question.

6. Fig5a, siPIAS4 should be validated by western blot.

Updated manuscript, See Extended Data Fig. 7g: Western blot validation of siRNA knock-down of PIAS4 has been performed and is included in extended figure 7g.

Minor points:

1. Subtitles should be provided in the 'MAIN' part.

Updated manuscript: This comment has been addressed in the revised manuscript

2. As Fig3c shows the degradation of WRN-Halo under 10 μ M WRNi for 16 h, how are the earlier images every few hours before it comes to 16h.

Updated manuscript: This is shown in multiple figures and described throughout the text. As described in the text, all degradation assays involving imaging were done for at least 16 h. Our model for how the inhibition of WRN results in chromatin trapping followed by SUMOylation, RNF4-mediated ubiquitylation and ultimately degradation by the proteasome is shown in Fig 6f.

3. In Fig5a,5b, why is WRN-Halo partially located in cytoplasm?

Updated manuscript, See Fig. 5a-b: Thank you for bringing this to our attention. This is non-specific staining of the JF dye that was used. We've replaced this image with cleaner staining conditions.

Reviewer #2 (Remarks to the Author):

WRN Inhibition Leads to its Chromatin-Associated Degradation Via the PIAS4-RNF4-p97/VCP Axis

Rodríguez Pérez et al.

Authors present work aimed at understanding the mechanism of action of a novel WRN helicase inhibitor. In brief the approach used involved cell biology and biochemical methods to characterise the effect of a small molecule WRNi on its target, WRN. In particular, the authors used advanced high-throughput imaging and analysis to extract information about the properties and behaviour of WRN under inhibition.

Major conclusions

The authors demonstrate that:

- WRNi is synthetic lethal in MSI-H cells, but not in MSS ones.
- WRNi leads to "trapping" of WRN on chromatin.
- Trapped WRN is destabilized and quickly degraded through an Ub-mediated pathway.
- WRN degradation follows the PIAS4-RNF4-p97 paradigm for the removal of chromatin trapped proteins.

Is the question addressed important?

MSI-H is a relatively common biomarker-defined cancer subgroup. The potential to target MSI-H cancers with WRNi is of a significant translational interest. As this work describes a novel WRNi it merits a significant interest.

Are the conclusions novel and will they influence thinking in the field?

The conclusions of the paper are not entirely novel as it was previously demonstrated that another small molecule WRNi inhibitor (NSC617145) can trap WRN on DNA. A number of other papers have shown WRN is synthetic lethal in MSI-H cells, and yet another set of papers have shown how chromatin-trapped proteins are extracted by the PIAS4-RNF4-p97 cascade. Hence, the authors have put all of these into a consistent model showing that the novel WRNi combines the known observation of synthetic lethality, trapping and chromatin extraction.

The authors have not provided any head-to-head comparisons with previously published inhibitors; neither have they assessed the in vivo potential of the new small molecule inhibitor, which would have been of a significant interest. Collectively, this undermines the potential impact of the work.

Quality of the data provided

The data provided in the manuscript is generally of a good quality. It is well-illustrated, analysed and communicated efficiently. Where it suffers is at certain points of clarity (detailed below) and consistency – showing the same set of experimental observations in different settings.

The data is communicated with sufficient detail in order to understand the main points of the experiment but some of the key underlying data (e.g. the siRNA screen data or the numerical data extracted from the image analysis) is not included and should be. The statistical analysis also lacks consistency of presentation, for example on bar plots sometimes individual values are included, but sometimes they are missing; the information on the number of reps and error bars could be improved.

Major points

1. In general, the novel WRNi should be characterised in a larger panel of cell lines, so that some comparison to the genetic dependency upon WRN in MSI-H tumor cell lines can be made. The authors have largely used HCT-116 cells, and to a degree U2-OS. They should demonstrate the effect of the drug on viability in a larger set of MSI-H cells (OVK18, OVK12, SW48, SNU1 etc.). Could the authors also compare this small molecule to NSC617145 in a head-to-head manner?

Updated manuscript, See Extended Data Fig. 1h-i and Extended Data Fig. 1l: We expanded our panel of MSI-H and MSS cancer cell lines and have now included that data in supplemental Fig1h & 1i. This new cell line data strengthens our original

observation as the six additional MSI-H cell lines all show decreased viability upon WRNi treatment while the additional three MSS cell line are unaffected by WRN inhibition. Additionally, we have included data profiling for NSC617145 and show that it is inactive across the assays used in this manuscript. This lack of WRN inhibition is consistent with other published findings for this compound, which have been described as screening artifacts (<https://doi.org/10.1002/cmdc.202300613>).

2. To address the specificity of the drug, besides the in vitro characterisation they provide, it would be important to test the drug in WRN KO models of which there are a few (MCF7, U2OS, HAP1). Obviously, these are not MSI-H cells (or else the KO will be synthetic lethal), but they still can provide information about this drug's specificity. For example WRN genetic inhibition causes PARP inhibitor sensitivity; does the small molecule inhibitor exacerbate PARPi sensitivity in WRN k/o cells or is it epistatic (the critical control here would be to show PARPi sensitivity with the inhibitor in WRN WT cells).

Updated manuscript: While we appreciate this line of investigation, we were unable to purchase or obtain these WRN KO cell lines so are unable to provide any additional data here. Further, given that we do not see the trapping in MSS lines such as U2OS, it is not clear what more we could learn from WRN KO in this context.

3. In vivo assessment of the effect of the new WRNi on MSI-H vs MSS cells is of a great importance.

Updated manuscript See Fig. 6a-d: There was universal interest among all three reviewers for in vivo data that supports our in vitro findings. To this end, we established an MSI-H xenograft model in mice using the human colorectal cancer cell line HCT-116. We then demonstrate that daily oral dosing of HR0761 results in a dose-dependent decrease in tumor growth in these animals. Western blot analysis of tumor samples from this study shows a dose-dependent decrease in WRN levels which is consistent with our in vitro findings that inhibited WRN is actively targeted for degradation.

4. Another general point about the data in all the biochemical figures is about the annotation of which cell lines are used. Sometimes these are HCT116 and sometimes their Halo-WRN HCT-116. This is important as it seems that the authors have tagged only one of the alleles (how many are they in HCT-116?), which may interfere with the interpretations of their single molecule results. They could at least discuss this point.

Updated manuscript: We tried to clarify this in the text of the revised manuscript to be very clear when we are using halo-tagged WRN cell lines versus untagged cells. In terms of the consequences of having a single WRN allele tagged in our HCT-116 line in the SMT experiments, we are only visualizing the halo-tagged WRN protein as we utilize a dye that covalently binds to Halo-tag for imaging. Due to the polyploid nature of

HCT-116 cells, we think it would be extremely difficult to try and tag every WRN allele in that cell background.

5. More importantly, from Extended Data Figure 2d it seems that WRNi degrades the Halo-WRN less efficiently than the untagged protein. Is this the case and how does this affect the rest of the results? On Figure 3a we see only a single band on the WB yet these are the HCT-116 Halo-WRN cells. Can the authors explain this discrepancy? Also they should provide full WB images for such experiments besides the cropped bands.

Updated manuscript: The reviewer is correct regarding the single band in the WRN Western show in Fig 3a as the figure legend was mislabeled indicating the use of WRN halo-tagged HCT-116 cells when parental HCT-116 cells were used. This has been corrected in the revised manuscript. With regards to whether halo-tagged WRN is degraded as efficiently as untagged WRN, we acknowledge that in Fig 2d there is slightly more halo-tagged WRN remaining after inhibitor treatment compared to untagged WRN. However, the overall reduction of both versions of WRN is extremely robust compared to the DMSO control bands. We do not think this is an issue, nor do we think this slight difference has any bearing on the experimental results reported in this manuscript. We are happy to provide full Western blots for all of our experiments.

6. The authors should demonstrate WRN trapping by conventional methods before basing all their conclusions on image-based analysis of a tagged protein. The standard assay for trapping is chromatin fractionation followed by WB. This data should be included in a revised Figure 2. Similarly, key statements about the removal of trapped WRN should be supported with biochemical data as well (chromatin fractionation), for example that data in Figure 4h-k, Figure 5i-j.

Updated manuscript, See Fig. 3q-i, Extended Data Fig. 4q-i, and Extended Data Fig. 5f. To provide additional validation of our SMT observations, we performed cell fractionation and chromatin precipitation experiments. We show that WRN protein remains bound to chromatin upon treatment with the WRN inhibitor in a dose-dependent manner. We also show that inhibition of VCP extends the timing where WRN can be detected by chromatin precipitation which is consistent with VCP's role in extracting "trapped" WRN from DNA. Lastly, as expected, we show that the binding of WRN to chromatin after inhibitor treatment is specific to MSI-H cancer cells as we don't observe this association after WRNi treatment in the MSS cell line U2OS.

7. The data on RNF4, PIAS4, p97 is convincing, but it could be further strengthened by the use of orthogonal approaches e.g. the use of dominant negative constructs, which do exist for RNF4 and p97. In addition, the data in figure 4 and 5 should be made consistent for each of the steps of the pathway. At the moment, there are mismatching pieces of information e.g. pH2AX for some, diffusion data for others.

Updated manuscript, See Fig. 4g-j and Extended Data Fig. 6 g: To address this comment, we utilized an inducible cellular expression system in combination with siRNA knock-down. We first show that in WRN inhibited cells, knock-down of RNF4 prevents WRN ubiquitylation and that this phenotype can be rescued by reintroducing wild-type RNF4 that is resistant to the siRNAs. We then show that overexpression of a catalytically dead version of RNF4 is unable to rescue ubiquitylation of inhibited WRN. We then demonstrate a direct interaction between WRN & RNF4 after inhibitor treatment using a TUBE pull-down approach. Lastly, we show that an inhibitor of VCP prolongs the interaction of WRN with chromatin due to inhibited VCP being unable to extract WRN for degradation. These data direct and conclusive evidence that RNF4 is indeed the ligase responsible for ubiquitylating inhibited WRN and that VCP plays an essential role in clearing trapped WRN from chromatin.

8. Of note, the data presented in Extended Data Figure 6e indicates that PIAS4 depletion has a very strong viability effect, which makes the interpretation of results from such cells unreliable.

Updated manuscript, See Extended Data Fig. 9e: We agree with the reviewer's mention of depletion of PIAS4 having a strong viability defect, therefore making data interpretation difficult from that experiment – siRNA depletions are difficult to titrate. This is why we decided to use a small molecule approach using a SUMO E1 inhibitor, which can be fine-tuned by titrating in the compound. As shown by our data, high concentrations of this inhibitor phenocopy what we see with PIAS4 depletion. However, at lower concentrations of compound used, synergy can be observed.

Minor points

Extended Figure 3C – have different loading controls and normalisations for HCT-116 and U2-OS. This needs to be made consistent.

Updated manuscript: This has been corrected in the revised manuscript.

Extended data 6f-h – please calculate Bliss coefficient for synergy/supra additivity.

Updated manuscript, See Extended Data Fig. 9e: This has been calculated as shown in extended data Fig 9d&e.

Reviewer #3 (Remarks to the Author):

The manuscript written by Pérez et al. studied the effect of WRN inhibition on WRN's dynamic localization within the nuclei of live cancer cells using single-molecule tracking method. The

results show that WRN inhibition traps the helicase on chromatin, requiring its degradation via the p97/VCP pathway, a process unique to MSI-H cells. The research identifies the PIAS4-RNF4 axis as crucial for WRN degradation and shows that simultaneous inhibition of WRN and SUMOylation is particularly toxic to MSI-H cells, suggesting a new regulatory mechanism for WRN that could lead to targeted cancer treatments.

I was asked to evaluate the single-molecule tracking experiments in this manuscript. Therefore, I will only comment on the SMT data and methodology. Overall, the SMT method used in this manuscript and the presented SMT data are sound. However, I have several concerns listed below:

1. The light sheet-based SMT platform needs to be more detailed described in this manuscript. Only mention “Oblique Line Scanning (OLS) illumination module” is not enough.

Updated manuscript, See Extended Data Fig. 3: We thank the reviewer for this comment, and we understand that we could have expanded on this. We have greatly expanded the Methods section to address this and provide details in Extended Data Fig. 3. We also invite the reviewer to look at our manuscript that was submitted to Nature Methods that goes into great detail regarding this OLS illumination modality. The preprint is in BiorXiv (Driouchi et al., 2023)

As general context, OLS microscopy offers a combination of field of view, confocality, illumination uniformity, and speed, that makes it very well suited to live-cell SMT studies. Through-objective illumination allows traditional sample mounting, a compact mechanical design and, as a result, suitability for large-scale and automated studies. These benefits are discussed and, where appropriate, quantified in the companion paper.

2. How many data points were used to calculate the diffusion coefficient in Figure 2b? There are many red lines in Figure 2b are straight line segments. Does this line only have two data points? What is the precision of the calculated diffusion coefficient in Figure 2b?
3. To validate the proposed SMT method, a control experiment needs to be performed. I suggest authors track 60 nm fluorescent beads in a water solution and calculate the diffusion coefficient of these beads to demonstrate the 6 $\mu\text{m}^2/\text{s}$ tracking speed.

Updated manuscript, See Extended Data Fig. 3: These are excellent questions. **Figure 2B** displays each track colored by the maximum likelihood estimate for its diffusion coefficient. In 2D, the max likelihood diffusion coefficient given a single trajectory is $\hat{D} = \frac{MSD}{4\Delta t}$ where MSD is the sum of squared displacements in the trajectory (in squared microns) and Δt is the frame interval (in seconds). This max likelihood estimate always exists for tracks with at least 2 points, but – as Reviewer #3 rightly points out – the estimate has higher dispersion/error for shorter tracks. Specifically, the Cramer-Rao lower bound for the max likelihood estimator is $CRLB(\hat{D}) \approx \frac{2D^2}{dL}$ where D is the true diffusion coefficient, d is the spatial dimension (here, $d=2$) and L is the number of jumps in the trajectory. (The approximation originates from neglecting the contribution of localization error; modeling localization error results in a more complex expression that we are happy to provide if

necessary.) This equation highlights that error in diffusion coefficient estimation will always be higher for faster tracks/larger D .

Due to this problem, estimating diffusion coefficient for very short, fast tracks is not a robust data processing strategy. Instead, the state array method (which is used to quantify SMT throughout the manuscript) integrates information over many trajectories to yield a more accurate estimate of diffusion coefficient than can be obtained from one trajectory alone. We include the visualization of the max likelihood tracks to highlight the character of the raw trajectories, including the presence of short, fast tracks that motivated our choice of processing strategy.

Importantly, we purposefully do **not** filter out short tracks throughout the manuscript. Such filtering introduces strong biases toward slow particles that remain in focus for many frames, creating systematically biased SMT results and reducing the assay's sensitivity to changes in fast free populations. These biases were investigated in detail by Hansen & Woringer et al. 2018, drawing on earlier work by Mazza et al. 2012 and Kues & Kubitscheck 2002.

Hansen & Woringer *et al. Elife* (2018). DOI: 10.7554/eLife.33125, PMID 29300163

Mazza D. *et al. Nucleic Acids Res.* (2012). DOI: 10.1093/nar/gks701, PMID 22844090

Kues T. & Kubitscheck U. *Single Molecules* (2002). [https://doi.org/10.1002/1438-5171\(200208\)3:4<218::AID-SIMO218>3.0.CO;2-C](https://doi.org/10.1002/1438-5171(200208)3:4<218::AID-SIMO218>3.0.CO;2-C).

Please see the BiorXiv pre-print for more details (Driouchi et al., 2023).

4. It is impressive the authors can extract the molecule's moving trajectory from blurry images (e.g., Figure 1a). It is necessary to provide the raw images data and present the outcome of each data process step. The data processing parameters also need to be provided.

Updated manuscript See *Extended Data Fig. 3*, *Supplementary Fig. 4*, and *Supplementary Mov. 1*:

We thank the reviewers for bringing this up, since the original manuscript had relatively sparse methods on data processing. The updated manuscript addresses this point in 4 ways:

1. The Methods for data processing parameters have been expanded to highlight each intermediate step of the single molecule tracking (SMT) pipeline, including particle detection, subpixel localization, and linking methods. This includes all hyperparameters used in the manuscript (such as detection thresholds, assumptions on PSF width, and search radii). Because most of these methods have been previously

described, we have also included references to primary sources throughout the updated Methods section.

2. We are providing a new supplementary figure with a schematic representation of the SMT data processing pipeline. This includes all of the steps necessary to produce the SMT-related figures in the manuscript, and complements the new mathematical description in the Methods.

3. We provide a new supplementary figure that compares the raw image data for the first and last frame of the SMT movie against the Hoechst channel and the final diffusion coefficient estimate. The goal is to enable the reader to compare the initial raw images against the final output shown in the main figures in the manuscript.

4. Finally, we are also including supplementary videos that show examples of SMT data used for deriving the results in the main text (Supplementary Fig. 4, and Supplementary Mov. 1)

5. What is the thickness of the light sheet? For a fast-moving molecule, for example, $6 \mu\text{m}^2/\text{s}$, it is very easy to diffuse out the excitation focal plane. How do you overcome this problem?

The OLS configuration used here matches that of (Driouchi et al., 2023), and so has an illumination pattern with a $1/e^2$ beam waist of approximately $3 \mu\text{m}$, inclined at a 60° angle with respect to the optic axis (giving a illumination width of approximately $6 \mu\text{m}$ when projected into the focal plane). The 1.27 numerical aperture, water-immersion, objective gives a depth of field on the order of 600nm . While the inclined and focused illumination pattern serves to reduce out-of-focus background contributions, the OLS depth of field is therefore limited by imaging, not illumination. Defocalization at the edges of the 600nm OLS depth of field is a significant limiting factor for observable track length in our experiments.

REVIEWER COMMENTS

Reviewer #1 (Remarks to the Author):

The authors have addressed the concerns of the manuscript, in particular as to the supplement of chromatin precipitation experiments and in vivo tests.

There are no further queries, only the extended figure 7e, 'Ei1' should be 'E1i'.

Reviewer #2 (Remarks to the Author):

WRN Inhibition Leads to its Chromatin-Associated Degradation Via the PIAS4-RNF4-p97/VCP Axis

Rodríguez Pérez et al.

Authors present work aimed at understanding the mechanism of action of a novel WRN helicase inhibitor. In brief the approach used involved cell biology and biochemical methods to characterise the effect of a small molecule WRNi on its target, WRN. In particular, the authors used advanced high-throughput imaging and analysis to extract information about the properties and behaviour of WRN under inhibition.

Major conclusions

The authors demonstrate that:

- WRNi is synthetic lethal in MSI-H cells, but not in MSS ones.
- WRNi leads to "trapping" of WRN on chromatin.
- Trapped WRN is destabilized and quickly degraded through an Ub-mediated pathway.
- WRN degradation follows the PIAS4-RNF4-p97 paradigm for the removal of chromatin trapped proteins.

Is the question addressed important?

MSI-H is a relatively common biomarker-defined cancer subgroup. The potential to target MSI-H cancers with WRNi is of a significant translational interest. As this work describes a novel WRNi it merits a significant interest.

Are the conclusions novel and will they influence thinking in the field?

The conclusions of the paper are not entirely novel as it was previously demonstrated that another small molecule WRNi inhibitor (NSC617145) can trap WRN on DNA. A number of other papers have shown WRN is synthetic lethal in MSI-H cells, and yet another set of papers have shown how chromatin-trapped proteins are extracted by the PIAS4-RNF4-p97 cascade. Hence, the authors have put all of these into a consistent model showing that the novel WRNi combines the known observation of synthetic lethality, trapping and chromatin extraction.

The authors have not provided any head-to-head comparisons with previously published inhibitors; neither have they assessed the in vivo potential of the new small molecule inhibitor, which would have been of a significant interest. Collectively, this undermines the potential impact of the work.

Quality of the data provided

The data provided in the manuscript is generally of a good quality. It is well-illustrated, analysed and communicated efficiently. Where it suffers is at certain points of clarity (detailed below) and consistency – showing the same set of experimental observations in different settings.

The data is communicated with sufficient detail in order to understand the main points of the experiment but some of the key underlying data (e.g. the siRNA screen data or the numerical data extracted from the image analysis) is not included and should be. The statistical analysis also lacks consistency of presentation, for example on bar plots sometimes individual values are included, but sometimes they are missing; the information on the number of reps and error bars could be

improved.

Major points

1. In general, the novel WRNi should be characterised in a larger panel of cell lines, so that some comparison to the genetic dependency upon WRN in MSI-H tumor cells lines can be made. The authors have largely used HCT-116 cells, and to a degree U2-OS. They should demonstrate the effect of the drug on viability in a larger set of MSI-H cells (OVK18, OVK12, SW48, SNU1 etc.). Could the authors also compare this small molecule to NSC617145 in a head-to-head manner?

Updated manuscript, See Extended Data Fig. 1h-i and Extended Data Fig. 1l: We expanded our panel of MSI-H and MSS cancer cell lines and have now included that data in supplemental Fig1h & 1i. This new cell line data strengthens our original observation as the six additional MSI-H cell lines all show decreased viability upon WRNi treatment while the additional three MSS cell line are unaffected by WRN inhibition. Additionally, we have included data profiling for NSC617145 and show that it is inactive across the assays used in this manuscript. This lack of WRN inhibition is consistent with other published findings for this compound, which have been described as screening artifacts (<https://doi.org/10.1002/cmdc.202300613>).

New reviewer comment: I would also encourage the authors to put new data in their point by point response – this would make it much simpler for reviewers to assess whether they have or have not addressed points made in the prior review and would respect the time reviewers dedicate, free of charge, to reviewing their manuscript.

The above point (the novel WRNi should be characterised in a larger panel of cell lines) has now been addressed.

2. To address the specificity of the drug, besides the in vitro characterisation they provide, it would be important to test the drug in WRN KO models of which there are a few (MCF7, U2OS, HAP1). Obviously, these are not MSI-H cells (or else the KO will be synthetic lethal), but they still can provide information about this drug's specificity. For example WRN genetic inhibition causes PARP inhibitor sensitivity; does the small molecule inhibitor exacerbate PARPi sensitivity in WRN k/o cells or is it epistatic (the critical control here would be to show PARPi sensitivity with the inhibitor in WRN WT cells).

Updated manuscript: While we appreciate this line of investigation, we were unable to purchase or obtain these WRN KO cell lines so are unable to provide any additional data here. Further, given that we do not see the trapping in MSS lines such as U2OS, it is not clear what more we could learn from WRN KO in this context.

New reviewer comment: The authors have not addressed this point, either experimentally, or by discussing this in the revised text of the manuscript.

3. In vivo assessment of the effect of the new WRNi on MSI-H vs MSS cells is of a great importance.

Updated manuscript See Fig. 6a-d: There was universal interest among all three reviewers for in vivo data that supports our in vitro findings. To this end, we established an MSI-H xenograft model in mice using the human colorectal cancer cell line HCT-116. We then demonstrate that daily oral dosing of HR0761 results in a dose-dependent decrease in tumor growth in these animals. Western blot analysis of tumor samples from this study shows a dose-dependent decrease in WRN levels which is consistent with our in vitro findings that inhibited WRN is actively targeted for degradation.

New reviewer comment: Importantly, the prior comment referred to "the effect of the new WRNi on MSI-H vs MSS cells". Whilst Fig 6 shows the effect in a MSI-H cell line xenograft, there is no assessment in mice carrying MSS xenografts. Lack of normal tissue tox somewhat assesses the effect in MSS cells, but this is only assessed as body weight. There is also no statistical analysis of the xenograft data.

4. Another general point about the data in all the biochemical figures is about the annotation of which cell lines are used. Sometimes these are HCT116 and sometimes their Halo-WRN HCT-116. This is important as it seems that the authors have tagged only one of the alleles (how many are they in HCT-116?), which may interfere with the interpretations of their single molecule results. They could at least discuss this point.

Updated manuscript: We tried to clarify this in the text of the revised manuscript to be very clear when we are using halo-tagged WRN cell lines versus untagged cells. In terms of the consequences of having a single WRN allele tagged in our HCT-116 line in the SMT experiments, we are only visualizing the halo-tagged WRN protein as we utilize a dye that covalently binds to Halo-tag for imaging. Due to the polyploid nature of HCT-116 cells, we think it would be extremely difficult to try and tag every WRN allele in that cell background.

New reviewer comment: HCT116 cells, like most MSI-H cells, have genomes that are relatively devoid of structural alterations and have few CN gains, amplifications etc.. i.e. the argument for not tagging all copies doesn't hold water. Plus, the reviewer simply asked authors to simply discuss this in the revised manuscript, which I cant find that they did.

5. More importantly, from Extended Data Figure 2d it seems that WRNi degrades the Halo-WRN less efficiently than the untagged protein. Is this the case and how does this affect the rest of the results? On Figure 3a we see only a single band on the WB yet these are the HCT-116 Halo-WRN cells. Can the authors explain this discrepancy? Also they should provide full WB images for such experiments besides the cropped bands.

Updated manuscript: The reviewer is correct regarding the single band in the WRN Western show in Fig 3a as the figure legend was mislabeled indicating the use of WRN halo-tagged HCT-116 cells when parental HCT-116 cells were used. This has been corrected in the revised manuscript. With regards to whether halo-tagged WRN is degraded as efficiently as untagged WRN, we acknowledge that in Fig 2d there is slightly more halo-tagged WRN remaining after inhibitor treatment compared to untagged WRN. However, the overall reduction of both versions of WRN is extremely robust compared to the DMSO control bands. We do not think this is an issue, nor do we think this slight difference has any bearing on the experimental results reported in this manuscript. We are happy to provide full Western blots for all of our experiments.

New reviewer comment: If the authors don't think this influences the conclusions of the study they might want to explain why in the discussion of the manuscript. The alternative is that readers see this difference, assume the authors have not noted this, and draw their own conclusions.

6. The authors should demonstrate WRN trapping by conventional methods before basing all their conclusions on image-based analysis of a tagged protein. The standard assay for trapping is chromatin fractionation followed by WB. This data should be included in a revised Figure 2. Similarly, key statements about the removal of trapped WRN should be supported with biochemical data as well (chromatin fractionation), for example that data in Figure 4h-k, Figure 5i-j.

Updated manuscript, See Fig. 3g-i, Extended Data Fig. 4g-i, and Extended Data Fig. 5f: To provide additional validation of our SMT observations, we performed cell fractionation and chromatin precipitation experiments. We show that WRN protein remains bound to chromatin upon treatment with the WRN inhibitor in a dose-dependent manner. We also show that inhibition of VCP extends the timing where WRN can be detected by chromatin precipitation which is consistent with VCP's role in extracting "trapped" WRN from DNA. Lastly, as expected, we show that the binding of WRN to chromatin after inhibitor treatment is specific to MSI-H cancer cells as we don't observe this association after WRNi treatment in the MSS cell line U2OS.

New reviewer comment: In Figure 4h it would help the reader if authros explained what concentrations of WRNi were used.

7. The data on RNF4, PIAS4, p97 is convincing, but it could be further strengthened by the use orthogonal approaches e.g. the use of dominant negative constructs, which do exist for RNF4 and p97. In addition, the data in figure 4 and 5 should be made consistent for each of the steps of the pathway. At the moment, there are mismatching pieces of information e.g. pH2AX for some, diffusion data for others.

Updated manuscript, See Fig. 4g-j and Extended Data Fig. 6 g: To address this comment, we utilized an inducible cellular expression system in combination with siRNA knock-down. We first show that in WRN inhibited cells, knock-down of RNF4 prevents WRN ubiquitylation and that this phenotype can be rescued by reintroducing wild-type RNF4 that is resistant to the siRNAs. We then show that overexpression of a catalytically dead version of RNF4 is unable to rescue ubiquitylation of inhibited WRN. We then demonstrate a direct interaction between WRN & RNF4 after inhibitor treatment using a TUBE pull-down approach. Lastly, we show that an inhibitor of VCP prolongs the interaction of WRN with chromatin due to inhibited VCP being unable to extract WRN for degradation. These data direct and conclusive evidence that RNF4 is indeed the ligase responsible for ubiquitylating inhibited WRN and that VCP plays an essential role in clearing trapped WRN from chromatin.

New reviewer comment: thank you for addressing this point.

8. Of note, the data presented in Extended Data Figure 6e indicates that PIAS4 depletion has a very strong viability effect, which makes the interpretation of results from such cells unreliable.

Updated manuscript, See Extended Data Fig. 9e: We agree with the reviewer's mention of depletion of PIAS4 having a strong viability defect, therefore making data interpretation difficult from that experiment – siRNA depletions are difficult to titrate. This is why we decided to use a small molecule approach using a SUMO E1 inhibitor, which can be fine-tuned by titrating in the compound. As shown by our data, high concentrations of this inhibitor phenocopy what we see with PIAS4 depletion. However, at lower concentrations of compound used, synergy can be observed.

New reviewer comment: authors might be wise to explain this in the revised main text of the manuscript.

Reviewer #3 (Remarks to the Author):

I still suggest the authors perform a control experiment to validate the diffusion coefficient, if possible. Tracking the 60 nm fluorescent beads water solution or tracking fluorescent dyes in glycerol-water mixtures, can be used to demonstrate the 6 $\mu\text{m}^2/\text{s}$ tracking speed of this method. Also, the error of the measured diffusion coefficient should be provided.

Reviewer #1 (Remarks to the Author):

The authors have addressed the concerns of the manuscript, in particular as to the supplement of chromatin precipitation experiments and in vivo tests.

There are no further queries, only the extended figure 7e, 'Ei1' should be 'E1i'.

Response to Reviewer: We thank the reviewer for thoroughly going over our revised manuscript. We have addressed the above comment and corrected the mistake.

Reviewer #2 (Remarks to the Author):

WRN Inhibition Leads to its Chromatin-Associated Degradation Via the PIAS4-RNF4-p97/VCP Axis

Rodríguez Pérez et al.

Authors present work aimed at understanding the mechanism of action of a novel WRN helicase inhibitor. In brief the approach used involved cell biology and biochemical methods to characterise the effect of a small molecule WRNi on its target, WRN. In particular, the authors used advanced high-throughput imaging and analysis to extract information about the properties and behaviour of WRN under inhibition.

Major conclusions

The authors demonstrate that:

- WRNi is synthetic lethal in MSI-H cells, but not in MSS ones.
- WRNi leads to “trapping” of WRN on chromatin.
- Trapped WRN is destabilized and quickly degraded through an Ub-mediated pathway.
- WRN degradation follows the PIAS4-RNF4-p97 paradigm for the removal of chromatin trapped proteins.

Is the question addressed important?

MSI-H is a relatively common biomarker-defined cancer subgroup. The potential to target MSI-H cancers with WRNi is of a significant translational interest. As this work describes a novel WRNi it merits a significant interest.

Are the conclusions novel and will they influence thinking in the field?

The conclusions of the paper are not entirely novel as it was previously demonstrated that another small molecule WRNi inhibitor (NSC617145) can trap WRN on DNA. A number of other papers have shown WRN is synthetic lethal in MSI-H cells, and yet another set of papers have shown how chromatin-trapped proteins are extracted by the PIAS4-RNF4-p97 cascade. Hence, the authors have put all of these into a consistent model showing that the novel WRNi combines the known observation of synthetic lethality, trapping and chromatin extraction.

The authors have not provided any head-to-head comparisons with previously published inhibitors; neither have they assessed the in vivo potential of the new small molecule inhibitor, which would have been of a significant interest. Collectively, this undermines the potential impact of the work.

Quality of the data provided

The data provided in the manuscript is generally of a good quality. It is well-illustrated, analysed and communicated efficiently. Where it suffers is at certain points of clarity (detailed below) and consistency – showing the same set of experimental observations in different settings.

The data is communicated with sufficient detail in order to understand the main points of the experiment but some of the key underlying data (e.g. the siRNA screen data or the numerical data extracted from the image analysis) is not included and should be. The statistical analysis also lacks consistency of presentation, for example on bar plots sometimes individual values are included, but sometimes they are missing; the information on the number of reps and error bars could be improved.

Major points

1. In general, the novel WRNi should be characterised in a larger panel of cell lines, so that some comparison to the genetic dependency upon WRN in MSI-H tumor cells lines can be made. The

authors have largely used HCT-116 cells, and to a degree U2-OS. They should demonstrate the effect of the drug on viability in a larger set of MSI-H cells (OVK18, OVK12, SW48, SNU1 etc.). Could the authors also compare this small molecule to NSC617145 in a head-to-head manner?

Updated manuscript, See Extended Data Fig. 1h-i and Extended Data Fig. 1l: We expanded our panel of MSI-H and MSS cancer cell lines and have now included that data in supplemental Fig1h & 1i. This new cell line data strengthens our original observation as the six additional MSI-H cell lines all show decreased viability upon WRNi treatment while the additional three MSS cell line are unaffected by WRN inhibition. Additionally, we have included data profiling for NSC617145 and show that it is inactive across the assays used in this manuscript. This lack of WRN inhibition is consistent with other published findings for this compound, which have been described as screening artifacts (<https://doi.org/10.1002/cmdc.202300613>).

New reviewer comment: I would also encourage the authors to put new data in their point by point response – this would make it much simpler for reviewers to assess whether they have or have not addressed points made in the prior review and would respect the time reviewers dedicate, free of charge, to reviewing their manuscript.

The above point (the novel WRNi should be characterised in a larger panel of cell lines) has now been addressed.

2. To address the specificity of the drug, besides the in vitro characterisation they provide, it would be important to test the drug in WRN KO models of which there are a few (MCF7, U2OS, HAP1). Obviously, these are not MSI-H cells (or else the KO will be synthetic lethal), but they still can provide information about this drug's specificity. For example WRN genetic inhibition causes PARP inhibitor sensitivity; does the small molecule inhibitor exacerbate PARPi sensitivity in WRN k/o cells or is it epistatic (the critical control here would be to show PARPi sensitivity with the inhibitor in WRN WT cells).

Updated manuscript: While we appreciate this line of investigation, we were unable to purchase or obtain these WRN KO cell lines so are unable to provide any additional data here. Further, given that we do not see the trapping in MSS lines such as U2OS, it is not clear what more we could learn from WRN KO in this context.

New reviewer comment: The authors have not addressed this point, either experimentally, or by discussing this in the revised text of the manuscript.

Response to Reviewer: We appreciate the intent of this comment and have addressed it to the best of our ability. Due to our inability to obtain a WRN KO cell lines, we instead generated an extensive data set showing biochemical, cellular and phenotypic characterization clearly demonstrating the specificity of this compound. Additionally, we believe that combination studies with PARP inhibitors are outside the scope of this paper as our work is focused on dissecting the biological process that result in WRN degradation in MSI-H cancer cell lines. As we state in our previous answer, given the plethora of data that we have obtained showing the specificity of this compound (i.e. biochemical, cellular, and phenotypic), we conclude that this compound is extremely specific.

3. In vivo assessment of the effect of the new WRNi on MSI-H vs MSS cells is of a great importance. Updated manuscript See Fig. 6a-d: There was universal interest among all three reviewers for in vivo data that supports our in vitro findings. To this end, we established an MSI-H xenograft model in mice using the human colorectal cancer cell line HCT-116. We then demonstrate that daily oral dosing of HR0761 results in a dose-dependent decrease in tumor growth in these animals. Western blot analysis of tumor samples from this study shows a dose-dependent decrease in WRN levels which is consistent with our in vitro findings that inhibited WRN is actively targeted for degradation.

New reviewer comment: Importantly, the prior comment referred to “the effect of the new WRNi on MSI-H vs MSS cells”. Whilst Fig 6 shows the effect in a MSI-H cell line xenograft, there is no assessment in mice carrying MSS xenografts. Lack of normal tissue tox somewhat assesses the effect in MSS cells, but this is only assessed as body weight. There is also no statistical analysis of the xenograft data.

Response to Reviewer: Thank you for your thoughtful feedback on this point. We appreciate your attention to detail and your valuable insights. We have included statistics on the *in vivo* mice measurements for Figure 6 (see screenshots below). We have also included details regarding the statistics that were ran in the figure legend (see screenshot below, highlighted in green).

Regarding your request for an MSS xenograft in our study, we respectfully submit that this falls beyond the scope of this work. Allow us to elaborate on our rationale:

Cell Viability Assessment:

- We conducted comprehensive cell viability assays across various MSS cell lines. Notably, no cell toxicity, or DNA damage, effects were observed in any of the MSS cells tested
- These findings reinforce the specificity of the WRN inhibitor and suggest no effects would be seen in a microsatellite stable (MSS) xenograft model

Weight Monitoring in Mice:

- As an additional measure, we monitored the weight of mice throughout the study. As the reviewer alludes to, the consistent weight maintenance in treated mice further supports the absence of normal cell toxicity by the WRN inhibitor.

Mechanism of Action (MoA) Clarification:

- Our primary objective was to mechanistically define the action of the WRN inhibitor. This manuscript delves into great detail to establish this, and we have unequivocally shown that inhibition of WRN leads to its chromatin trapping, resulting in degradation by the SUMO/RNF4/VCP/Proteasome axis.

In light of these compelling data, we believe that the inclusion of an MSS mice xenograft would not substantially enhance the scientific rigor or relevance of our study, nor would it change the main conclusions of our work, which was to establish a link between the WRN inhibition and WRN degradation, and the mechanisms that govern this.

a

b

extracted from chromatin by p97/VCP, leading to its degradation by the proteasome. For a and b, significance was assessed via a 2-way ANOVA with Dunnett's multiple comparison test, with a $P > 0.01$. At the terminal Day 18 timepoint, all weight change means were ns from control. For the terminal tumor volume means, dosing groups were compared against the vehicle control (**** = $P < 0.0001$).

4. Another general point about the data in all the biochemical figures is about the annotation of which cell lines are used. Sometimes these are HCT116 and sometimes their Halo-WRN HCT-116. This is important as it seems that the authors have tagged only one of the alleles (how many are they in HCT-116?), which may interfere with the interpretations of their single molecule results. They could at least discuss this point.

Updated manuscript: We tried to clarify this in the text of the revised manuscript to be very clear when we are using halo-tagged WRN cell lines versus untagged cells. In terms of the consequences of having a single WRN allele tagged in our HCT-116 line in the SMT experiments, we are only visualizing the halo-tagged WRN protein as we utilize a dye that covalently binds to Halo-tag for imaging. Due to the polyploid nature of HCT-116 cells, we think it would be extremely difficult to try and tag every WRN allele in that cell background.

New reviewer comment: HCT116 cells, like most MSI-H cells, have genomes that are relatively devoid of structural alterations and have few CN gains, amplifications etc.. i.e. the argument for not tagging all copies doesn't hold water. Plus, the reviewer simply asked authors to simply discuss this in the revised manuscript, which I cant find that they did.

Response to Reviewer: Thank you for your thoroughness. As stated in our previous response to the reviewer, we have clearly indicated what cell lines were used for each experiment, and we hope this alleviates any confusion.

Regarding the heterogenous tagging of WRN in HCT116 cells, the method of single molecule tracking (SMT) visualizes only the HaloTagged protein of interest. Given this nature of the technique, having a heterozygous population will not impact the interpretations/results of this experiment. Furthermore, we have performed orthogonal assays to validate the observations we see in SMT. Lastly, we have added some additional commentary around this point in the updated manuscript and hope this finally addresses these concerns.

5. More importantly, from Extended Data Figure 2d it seems that WRN_i degrades the Halo-WRN less efficiently than the untagged protein. Is this the case and how does this affect the rest of the results? On Figure 3a we see only a single band on the WB yet these are the HCT-116 Halo-WRN cells. Can the authors explain this discrepancy? Also they should provide full WB images for such experiments besides the cropped bands.

Updated manuscript: The reviewer is correct regarding the single band in the WRN Western show in Fig 3a as the figure legend was mislabeled indicating the use of WRN halo-tagged HCT-116 cells when parental HCT-116 cells were used. This has been corrected in the revised manuscript. With regards to whether halo-tagged WRN is degraded as efficiently as untagged WRN, we acknowledge that in Fig 2d there is slightly more halo-tagged WRN remaining after inhibitor treatment compared to untagged WRN. However, the overall reduction of both versions of WRN is extremely robust compared to the DMSO control bands. We do not think this is an issue, nor do we think this slight difference has any bearing on the experimental results reported in this manuscript. We are happy to provide full Western blots for all of our experiments.

New reviewer comment: If the authors don't think this influences the conclusions of the study they

might want to explain why in the discussion of the manuscript. The alternative is that readers see this difference, assume the authors have not noted this, and draw their own conclusions.

Revised Manuscript: We appreciate the thoroughness of the reviewer in identifying these details. The small change referred to in Extended Data Fig 2. does not change the conclusions of the work. Orthogonal assays that were performed using immunofluorescence, biochemical, or SMT methods all point to the same conclusion: WRN is degraded by the SUMO/RNF4/VCP/Proteasome axis. As we previously stated, we do not think this is an issue or that this slight difference has any bearing on the experimental results or overall conclusions reported in this manuscript. Addressing this in the text would detract/distract the reader from the main work we present.

6. The authors should demonstrate WRN trapping by conventional methods before basing all their conclusions on image-based analysis of a tagged protein. The standard assay for trapping is chromatin fractionation followed by WB. This data should be included in a revised Figure 2. Similarly, key statements about the removal of trapped WRN should be supported with biochemical data as well (chromatin fractionation), for example that data in Figure 4h-k, Figure 5i-j. Updated manuscript, See Fig. 3g-i, Extended Data Fig. 4g-i, and Extended Data Fig. 5f: To provide additional validation of our SMT observations, we performed cell fractionation and chromatin precipitation experiments. We show that WRN protein remains bound to chromatin upon treatment with the WRN inhibitor in a dose-dependent manner. We also show that inhibition of VCP extends the timing where WRN can be detected by chromatin precipitation which is consistent with VCP's role in extracting "trapped" WRN from DNA. Lastly, as expected, we show that the binding of WRN to chromatin after inhibitor treatment is specific to MSI-H cancer cells as we don't observe this association after WRNi treatment in the MSS cell line U2OS.

New reviewer comment: In Figure 4h it would help the reader if authors explained what concentrations of WRNi were used.

Revised Manuscript: Thank you for calling this out. We have included the concentrations of inhibitor used in the figure legend (see screenshot below, purple highlighter).

but are not observed when treated with siRNF4 oligos. **g.** WRN degradation upon its inhibition can be functionally rescued. Western blot analysis of HCT-116^{WRN-Halo;RNF4} cells treated with 10 μ M WRNi after depletion of RNF4, and subsequent induction of stably integrated RNF4. **h.** The degradation of WRN after its inhibition is dependent on the ubiquitin ligase activity of RNF4. Western blot analysis as in **g**, but using an RNF4^{C132S/C135S} catalytic mutant. **i.** Functional rescue of WRN degradation can be visualized via

7. The data on RNF4, PIAS4, p97 is convincing, but it could be further strengthened by the use of orthogonal approaches e.g. the use of dominant negative constructs, which do exist for RNF4 and p97. In addition, the data in figure 4 and 5 should be made consistent for each of the steps of the pathway. At the moment, there are mismatching pieces of information e.g. pH2AX for some, diffusion data for others.

Updated manuscript, See Fig. 4g-j and Extended Data Fig. 6 g: To address this comment, we

utilized an inducible cellular expression system in combination with siRNA knock-down. We first show that in WRN inhibited cells, knock-down of RNF4 prevents WRN ubiquitylation and that this phenotype can be rescued by reintroducing wild-type RNF4 that is resistant to the siRNAs. We then show that overexpression of a catalytically dead version of RNF4 is unable to rescue ubiquitylation of inhibited WRN. We then demonstrate a direct interaction between WRN & RNF4 after inhibitor treatment using a TUBE pull-down approach. Lastly, we show that an inhibitor of VCP prolongs the interaction of WRN with chromatin due to inhibited VCP being unable to extract WRN for degradation. These data direct and conclusive evidence that RNF4 is indeed the ligase responsible for ubiquitylating inhibited WRN and that VCP plays an essential role in clearing trapped WRN from chromatin.

New reviewer comment: thank you for addressing this point.

8. Of note, the data presented in Extended Data Figure 6e indicates that PIAS4 depletion has a very strong viability effect, which makes the interpretation of results from such cells unreliable.

Updated manuscript, See Extended Data Fig. 9e: We agree with the reviewer's mention of depletion of PIAS4 having a strong viability defect, therefore making data interpretation difficult from that experiment – siRNA depletions are difficult to titrate. This is why we decided to use a small molecule approach using a SUMO E1 inhibitor, which can be fine-tuned by titrating in the compound. As shown by our data, high concentrations of this inhibitor phenocopy what we see with PIAS4 depletion. However, at lower concentrations of compound used, synergy can be observed.

New reviewer comment: authors might be wise to explain this in the revised main text of the manuscript.

Revised Manuscript: This is a very good suggestion, and we have incorporated this into the main text (see screenshot below).

after depletion of RNF4 (Extended Data Fig. 9c). Strikingly, depletion of PIAS4 led to a profound viability defect even in the absence of WRNi. We reason this is due to SUMOylation being a critical process for the maintenance of DNA homeostasis, and overall cell health, making chronic depletion of the SUMO ligase PIAS4 is not well tolerated by cells. To circumvent this, we used a small molecule inhibitor (SUMOi) in combination with WRNi. This treatment results in synergistic shift in the EC₅₀ of WRNi (Fig. 5k and Extended Data Fig. 9d-f) (lanevski et al., 2022). Many other areas of potential intervention exist, such as

Reviewer #3 (Remarks to the Author):

I still suggest the authors perform a control experiment to validate the diffusion coefficient, if possible. Tracking the 60 nm fluorescent beads water solution or tracking fluorescent dyes in glycerol-water mixtures, can be used to demonstrate the 6 $\mu\text{m}^2/\text{s}$ tracking speed of this method. Also, the error of the measured diffusion coefficient should be provided.

Response to Reviewer: This comment is addressed in exquisite detail in a methods paper that is currently under review (and in BiorXiv; see screenshot below of the OLS manuscript).

Regarding the error of diff. co. measurements, we believe we have dot plots when we perform measurements, but if the reviewer is referring to a specific example, we'll be happy to address this.

REVIEWERS' COMMENTS

Reviewer #2 (Remarks to the Author):

Although the authors have still not addressed all of the points raised in the review, I would not want to stand in the way of this being published. I would though suggest one thing - where they disagree with a reviewers comments, it might make more sense to highlight the point in the Discussion and provide their argument against the point. This would be more scientific than simply dismissing points and/or arguing things are outside scope (if you think an experiment is outside scope, point this out in the revised discussion).

Reviewer #3 (Remarks to the Author):

The authors have addressed all my concerns about the manuscript. I have no further questions and would be happy to recommend the publication of this manuscript.